



# Importance of spatial and depth-dependent drivers in groundwater level modeling through machine learning

Pragnaditya Malakar[1], Abhijit Mukherjee[1,2,3], Soumendra N. Bhanja[4], Dipankar Saha[5], Ranjan Kumar Ray[6], Sudeshna Sarkar[7], Anwar Zahid[8]

[1]Department of Geology and Geophysics, Indian Institute of Technology Kharagpur, West Bengal 721302, India
[2]School of Environmental Science and Engineering, Indian Institute of Technology Kharagpur, West Bengal 721302, India
[3]Applied Policy Advisory for Hydrogeoscience (APAH) Group, Indian Institute of Technology Kharagpur, West Bengal 721302, India
[4]Interdisciplinary Centre for Water Research, Indian Institute of Science, Bangalore, Karnataka 560054, India
[5]Formerly Central Ground Water Board, Ministry of Water Resources, River Development and Ganga Rejuvenation, Government of India, Faridabad, Haryana, India
[6]Central Ground Water Board (CGWB), Bhujal Bhawan, NH-IV, Faridabad, India
[7]Department of Computer Science and Engineering, Indian Institute of Technology Kharagpur, West Bengal 721302, India
[8]Bangladesh Water Development Board (BWDB), Dhaka, Bangladesh

*Correspondence:* Pragnaditya Malakar (pragnadityamalakar@gmail.com) and Abhijit Mukherjee (amukh2@gmail.com)

## Abstract

The water and food security of South Asia is embedded in the groundwater resources of the transboundary aquifer system of Indus-Ganges-Brahmaputra-Meghna (IGBM) rivers, which has been subjected to diverse natural and anthropogenic triggers. Thus, understanding the relative importance of such triggers in groundwater level change and developing a prediction framework is essential to sustain future stress. Although a number of studies on groundwater level prediction and simulation exist in the literature, characterization of predictive performances of groundwater level modeling using a large network of ground-based observations ($n = 2303$) is not yet reported. To identify the spatial and depth-wise predictors influence, here, we used linear regression based dominance analysis and machine learning methods (Support Vector Machine and Artificial Neural network) on long term (1985-2015) GWLs and/or climatic variables in the parts of IGBM basin aquifers. The results from the dominance analysis show that groundwater level change is primarily influenced by abstraction and population in most of the IGBM, whereas in the Brahmaputra basin, precipitation exhibits greater influence. Our results show a large proportion of the observation wells ($n >50\%$ for ANN and $n >65\%$ for SVM) demonstrate good correlation ($r> 0.6, p<0.05$), Nash-Sutcliff efficiency ($NSE >0.65$), and normalized root mean square error ($RMSE_n<0.6$) between the observed and simulated values. However, the results in the highly abstracted parts of the basin are poor, due to insufficient knowledge of groundwater abstraction. Furthermore, a significant decrease in performance from shallow (intake depth $< 35m$) to deep observation wells (intake depth $> 35m$) could be linked to the change in groundwater abstraction pattern from shallow to deep groundwater in recent times. We also find that, in areas where natural factors dominate over anthropogenic factors, climatic variables may be used as suitable predictors for the groundwater level.

## 1 Introduction

Groundwater is the largest accessible storage of global freshwater resources, which sustains most of the human consumption, including the global irrigational water supply and acts as an inventory in times of droughts (Taylor et al., 2013; Famiglietti, 2014). However, in the past few decades, most of the major aquifers around the world are experiencing significant depletion in groundwater storage related to the increasing agricultural productivity for the growing population (Siebert et al., 2015).




The transboundary aquifer system of South Asia is known as the Indus-Ganges-Brahmaputra-Meghna (IGBM) basin aquifer. The IGBM is one of the hotspots regarding global water and food security having ~114 million ha (>50% of total area) of net cropping area (Mukherjee et al., 2015; Sharma et al., 2008). The regional precipitation, mainly controlled by the Asian summer monsoon from June to September (Lutz et al., 2019), significantly increases the groundwater storage (Singh et al., 2019) in the region.

However, the irrigational water demand in this region is approximately 280 km$^3$/yr (Tiwari et al., 2009), which accounts ~25% global groundwater abstraction (MacDonald et al., 2016). As a result of the pervasive groundwater withdrawals, IGBM experiences rapid groundwater depletion, predominantly in North-West India, South-East India (Bengal basin), and the Meghna basin in Bangladesh (MacDonald et al., 2016). These densely populated agricultural region of IGBM is dependent on the groundwater-fed irrigation for crop production, primarily for the summer (Rabi) and winter (Kharif) crops. In addition to the water

crisis in terms of groundwater quantity (Mukherjee et al., 2007; Rodell et al., 2009; MacDonald et al., 2015; Bhanja et al., 2017a, 2017b; Mukherjee et al., 2018; Bhanja and Mukherjee, 2019, Bhanja et al., 2019a), groundwater quality is also a major issue, due to the presence of geogenic arsenic, salt and fluoride contamination (Mukherjee et al., 2015; MacDonald et al., 2016, Podgorski et al., 2018). Thus, posing a severe threat to water sustainability for the millions of people in South-Asia. Hence, modeling groundwater resources is crucial for sustaining a balance between groundwater supply and demand for the large population.

Over the years, the simplistic approach and acceptable results of the machine learning (ML) methods are preferred when the underlying physical system is not well understood. GWL modeling based on ML has the unique ability to find the likely relationships between GWL and controlling hydro-climatic-anthropogenic variables without constructing knowledge-driven conceptual or physically-based models. Therefore, researchers have studied the performance of ML methods for GWL modeling in India and Bangladesh (Nayak et al., 2006; Nury et al., 2017; Malakar et al., 2018; Mukherjee and Ramachandran, 2018; Bhanja

et al., 2019b; Sun et al., 2019; Yadav et al., 2019 and the references therein) and other parts of the world (Coulibaly et al., 2001; Feng et al., 2008; Sun, 2013; Nourani and Mousavi, 2016; Sun et al., 2016; Yoon et al., 2016; Barzegar et al., 2017; Ebrahimi and Rajaee, 2017; Wunsch et al., 2018; Chen et al., 2019; Lee et al., 2019 and the references therein). Most of these studies used popular methods like Artificial Neural Network (ANN), hybrid-ANN, Adaptive neuro-fuzzy inference system (ANFIS), Support Vector Machine (SVM) and few others using a wide range of frequency and temporal data on past GWLs, satellite observations

derived groundwater storage (GWS), Normalized difference vegetation index (NDVI)), meteorological variables, river discharge, variables on groundwater use and few dummy variables to simulate and/or predict GWLs. A recent study by Mukherjee and Ramachandran (2018) simulated GWLs for a small number ($n = 5$) of in-situ observation wells in India using Linear Regression Model (LRM), Artificial Neural Network (ANN) and Support Vector Regression (SVR) with Gravity Recovery and Climate Experiment (GRACE) derived terrestrial water storage (TWS) change and meteorological variables. However, these previous

studies, including studies on India and Bangladesh, are mainly small-scale studies, and due to the small number of observation wells, they are unable to characterize the spatial variability in model performances extensively. Furthermore, the temporal extent of the studies on India and Bangladesh is often short (e.g., Mukherjee and Ramachandran (2018) considered the time period from 2005 to 2018). Hence the predictions are based on the short-term trends of dependent variables and do not consider the long-term variability. Moreover, using a combination of physically-based modeling and deep convolutional neural network (CNN), Sun et

al. (2019) matched the GRACE based and simulated (by a land surface model as inputs) terrestrial water storage anomalies (TWSA). They further compared the calculated in-situ GWS (using specific yields and in-situ GWLs) with the variation between the observed and simulated model values. However, this study does not use in-situ GWLs as model input and mainly based on the satellite observations and land surface model outputs. Moreover, a recent study discussed the significant impact of population growth in GWL estimation and prediction in urban areas in India (Yadav et al., 2019).



Central Ground Water Board (CGWB, Government of India) and Bangladesh Water Development Board (BWDB, Government of Bangladesh) cover a dense network of groundwater level measurement locations across India (n~13500, in the IGBM basin) and Bangladesh (n~1300), respectively (CGWB, 2014; Shamsudduha et al., 2011). Although the GWL monitoring locations could be unevenly spaced, it provides a spatially high-resolution monitoring system when compared to the other major alternatives (e.g., satellite observations) and also shows low spatial error (Bhanja et al., 2017b). The variation of GWLs, measured in observation

wells, provides critical information on the groundwater development stage and aquifer dynamics of the region. Hence, the existing GWL time-series data from the large observation network could be effectively used to understand the groundwater system and resource prediction.

The objective of the study is to understand a) the spatial and depth-wise predictive performance of groundwater levels across the vast, heterogeneous aquifer systems of the IGBM using machine learning methods (ANN and SVM), b) evaluate the importance

of major natural and anthropogenic factors in groundwater level change using Linear Regression based dominance analysis, and c) delineate the most robust and realistic approach for using machine learning to simulate groundwater levels. This study intends to bridge the gaps between limitations of some previous literature (e.g., Mukherjee and Ramachandran, 2018; Sun et al., 2019, discussed above), as well as try to explore the limit/s and capabilities of the aforesaid computational methods in modeling groundwater levels. The originality of the article lies in addressing some critical aspects. Firstly, to understand the spatial variability

in machine learning-based model performances, we have considered a large network of monitoring wells (n = 2303) from 1985 to 2015, to simulate GWLs in the IGBM. Secondly, considering the variable pattern of groundwater abstraction, we showed the significance of well depth (intake depth of the observation wells) information in GWL modeling using machine learning. Thirdly, we used meteorological variables exclusively to simulate in-situ GWL. Fourthly, based on dominance analysis and outputs from the machine learning models, we investigated the basin specific predictor (both natural and human-induced) importance in GWL

modeling.

## 2 Material and methods

### 2.1 Study area

The Indus-Ganges-Brahmaputra-Meghna (IGBM) alluvial aquifer system was formed with the eroded Himalayan sediments, distributed by the main rivers in the region, i.e., Indus, Ganges, and Brahmaputra (Bonsor et al., 2017). The IGBM basin supports

one of the world's most important high yielding transboundary aquifer system that expands across the fertile plains of Pakistan, India, Bangladesh, and Nepal (Mukherjee et al., 2015). In this study, we have considered only the Indian and Bangladesh part of the IGBM basin based on data availability (Figure 1a). Based on the location of the rivers, the study area is subdivided into four sub-basins, i.e., the Brahmaputra basin (B), Meghna basin (M) in the east, the Indus basin (I) in the west, and the Ganges basin (G) in the middle (Figure 1a). The IGBM aquifer system is extremely heterogeneous in terms of groundwater recharge, groundwater

abstraction pattern, climatic, and subsurface hydraulic properties (Mukherjee et al., 2015; Bonsor et al., 2017). The annual precipitation in the basin has a declining gradient from east to west, with the highest precipitation observed in the Meghna and Brahmaputra basin and lowest observed in the Indus basin (Table S1, Figure 1d). Most of the basin suffers from high groundwater abstraction. The Indus and Meghna basin have the highest rate of groundwater abstraction in the basin, followed by the Ganges and Brahmaputra Basin (Table S2, Figure 1c). Long-term average map of other climatic factors (e.g., temperature and potential

evapotranspiration) for the years 1985 to 2015 have been shown in Figure S1.

### 2.2 Data

### 2.2.1 In-situ groundwater level measurements





Long-term groundwater level (GWL) data from ~13500 observation wells (initial well number, before filtering and post-processing) for 31 years (1985 – 2015) for India and Bangladesh were retrieved from the CGWB and BWDB, respectively (Table
S3). The GWL data for India are available four times a year, i.e., for January (late post-monsoon), May (pre-monsoon), August (monsoon), and November (early post-monsoon) (CGWB, 2014). However, for Bangladesh, GWL data is available in a weekly format. The GWL data for Bangladesh are then transformed (using averaging the weekly data) into the quarterly structure to maintain uniformity with the CGWB dataset.

### 2.2.2 Climate Variables

In this study, we used gridded climate data that include daily and/or monthly precipitation, daily and/or monthly temperature (maximum, minimum, and mean) and monthly potential evapotranspiration for 1985 to 2015. This data have been collected from various governmental and other sources (Table S3).

### 2.2.3 Groundwater withdrawals

We have calculated the basin wise annual groundwater withdrawals estimates for the basin and sub-basins by using already
published estimates and combined them with the irrigation well numbers, pumping rate datasets (Mukherjee et al., 2007; Bhanja et al., 2017a) from various sources (Table S3). Groundwater withdrawals data for India were retrieved from Dynamic Ground Water Resources of India (CGWB, 2019) and Food and Agriculture Organization of the United Nations (AQUASTAT, 2018); pumping wells statistics were used from Minor irrigation census (Minor irrigation, 2017). For Bangladesh, the groundwater withdrawals data were derived by integrating data from local and published datasets (AQUASTAT, 2018; Bangladesh Agricultural
Development Corporation, 2017) (Table S3).

### 2.2.4 Population

Gridded Population count data (spatial resolution: 2.5 arc-minute resolution, ~ 5 km grids) of the World, Version 4 (GPWv4) ( (NASA Socioeconomic Data and Applications Center (SEDAC), 2018) were used in the study.

### 2.3 Data management, selection criteria, and classification

The groundwater level data used in the study contains inconsistencies and outliers. Firstly, to remove outliers, Tukey's fence approach (Tukey, 1977) was applied to the datasets. To attain temporal consistency, the data were further filtered with the selection criteria that include the observation wells having three out of four data in each year for all 31 years. The usable number of observation wells (Figure S2) was reduced significantly (from $n=13465$ to usable $n=2303$), following the application of these filters and data processing. The missing values in the GWL time series data were filled using Multiple imputation (Azur et al.,
2011; Resche-Rigon and White, 2018) with MICE (Van Buuren and Groothuis-Oudshoorn, 2011; Gibrilla et al., 2018) package in R statistical software. In order to allow the variables to get equal consideration in the models, all the time series data were normalized ($x_{norm}$) using maximum ($x_{max}$) and minimum ($x_{min}$).

$$x_{norm} = \frac{x - x_{min}}{x_{max} - x_{min}} \qquad (1)$$

The observation wells in the study area were classified by their intake depth: SH (*shallow observation wells; intake depth < 35 m; n = 2080*) and DP (*deep observation wells; intake depth >35 m; n = 223*). These observation well classifications were made based





on the dominant subsurface depth of the irrigation wells in the region (Minor irrigation, 2017). The summary of the observation well numbers is given in Table S4.

**2.4 Dominance analysis**

The relative importance of the predictor variables is determined by the dominance analysis, which computes the coefficient of determination ($R^2$) in multiple regression (Azen and Budescu, 2006; Budescu, 1993; Thomas and Famiglietti, 2019). Yearly precipitation, temperature, groundwater withdrawals, population, and potential evapotranspiration for the IGBM and each of sub-basins were taken as the independent variable to understand their relationship with the dependent variable GWLs. Here, the conditional dominance of the variables for *p-1* sub-models is performed (where *p* is the numeric value of total sub-models) (Thomas

and Famiglietti, 2019). A comprehensive narrative on the dominance analysis can be found in Budescu (1993) and Azen and Budescu (2006).

**2.5 Multi-model analysis**

In this study, machine learning (ML) based methods have been used to study the efficiency and effect of major influencing variables in simulating GWL in the IGBM. Two widely used methods applied in the study are ANN and SVM.

**2.5.1 Artificial Neural Network (ANN)**

ANN is a data-driven computational method, which follows the biological neural system. ANN are widely used to establish the functional relationships between variables and to simulate and predict values based on historical time series values.

$$Y_m = f(X_n) \tag{2}$$

Here, $X_i$ is a n-dimensional input variable and $Y_j$ is the m dimensional output variable; ($i = 1, 2, ......., n$) , ($j = 1, 2, ............, m$).

An ANN is comprised of processing elements called neurons and a connection network linking the neurons. Generally, an ANN structure has three separate layers: the Input layer that consists of the input variable; the Hidden layer(s) where the data processing is executed; and the results are generated in the output layer. The results in the output layer are produced through an activation (transfer) function, which uses the biased and weighted input carried by each neuron. The ANNs are trained with data to tune the weights and biases so that the model performances can be optimized. Here we developed feed-forward neural networks (FNN)

(Svozil et al., 1997), which is a type of ANN that propagate the input signal from the input layer to the output layer in a forward direction. The FNN is trained with the Levenberg-Marquardt (LM) algorithm since this combination is reported to be efficient, stable and less affected by local minima (Nayak et al., 2006; Krishna et al., 2008; Nourani et al., 2008; Chang et al., 2015; Rajaee et al., 2019)

An FNN can be defined as

$$y_k = S_1 \left( \sum_{j=1}^{J} w_j S_2 \left( \sum_{i=1}^{I} w_i x_i \right) \right) \tag{3}$$

Where $x_i$ and $y_k$ are the input and output vector respectively; $S_1$ and $S_2$ are the transfer (activation) functions; the weights of the nodes in the input- hidden-layer are $w_i$, and the weights of the nodes in the output layer are $w_j$. In this study, we used the Logistic sigmoidal transfer function, which is the most frequently used transfer function for ANN in GWL modeling, since the function is continuous, monotonically increasing, and differentiable (Ravansalar and Rajaee, 2015).





The logistic sigmoidal transfer function may be expressed as,

$$s(x) = \frac{1}{1 + e^{-x}} \tag{4}$$

### 2.5.2 Support Vector Machine (SVM)

SVM is a non-parametric machine learning method based on the statistical learning theory (Cortes and Vapnik, 1995). In SVM, the model structure is designed by a training process and not priori determined. The general concept of SVM is to separate classes

in a classification problem by identifying hyperplanes. SVM transforms the nonlinear decision boundaries in the original space to the linear decision boundaries in the new infinite-dimensional space with a kernel function (Cortes and Vapnik, 1995). The "support vectors" are the points that lie closest to the optimal separating hyperplane. For the regression problem, the support vector regression (SVR) method is used, which applies similar principles to SVM. SVM adopts the structural risk minimization hypothesis and dimension theory of Vapnik to attain an enhanced generalization capability (Cortes and Vapnik, 1995).

Assuming, there are $N$ samples $\{x_k;\ y_k\}^N_{k=1}$; $x$ is the input vector of $m$ components, and $y$ is the output vector of $n$ component, i.e., $x \in R_m; y \in R_n$.

The Support vector regression (SVR) function (*f*) is expressed as

$$f(x) = w.\phi(x) + b \tag{5}$$

Here $\emptyset$ is a nonlinear transfer function, which maps $x_k$ into the feature space, where a simple linear regression can be performed.

The weight vector is *w*, and the bias is *b* (Yoon et al., 2011). A detailed overview of SVM can be found in Vapnik (1998).

In this study for training the models, the radial basis function (RBF) kernel has been used. The RBF function has been chosen as the kernel function because of its reliable performance and accuracy (Suryanarayana et al., 2014; Rajaee et al., 2019). Five-Fold cross-validation has been applied to achieve better results. For building the SVM models, the R package 'e1071' (Meyer et al., 2019) was used. The training parameters (cost, gamma, and epsilon) have been tuned using the 'tune' function of the e1071

package.

### 2.5.3 Model building

In this study, three different types of models for each ANN and SVM were developed for simulating the in-situ GWLs. In the first model (Model A), GWL (as an independent variable) is used to simulate GWL (the dependent variable). In the second model (Model B), in addition to GWLs, climatic variables (i.e., precipitation, temperature (maximum, minimum, mean), and potential

evapotranspiration) were incorporated in the model (as independent variables) to simulate GWLs. In the third model (Model C), only climatic variables were used in the model as independent variables to simulate GWLs. However, Due to the paucity of spatio-temporal data, abstraction and population could not be included as an explanatory variable in the machine learning modeling.

### 2.5.4 Model training and test data set development

The total dataset of 31 years (*31 × 4 = 124 seasons, 1985 - 2015*) is subdivided into training and testing datasets. For each well

location, we have created training and testing data set for five different configurations (Table S5). Configuration – 4 was selected for final modeling. This configuration is adapted, maintaining a trade-off between the model performances and the highest possible test set. All the final models (SVM Model A, B, C; ANN Model A, B, C) have been trained for each of the 2303 in-situ wells, using data for 84 seasons (*21 years, 1985-2005, training period, 68% of the total time period*). Then the simulated GWLs for the





next 40 seasons (*10 years, 2006-2015, test period, 32% of the total time period*) have been compared with the observed GWLs.
The training and testing of the models have been accomplished with entire temporal datasets for each of the 2303 well locations.
The intake depth information of the observation wells was not provided in the models.

### 2.5.5 Model performance comparison

The correlation coefficient, Nash-Sutcliff efficiency, Normalized root mean square error, Standard error, and Percentage prediction
error are applied to evaluate the performance of the models. Normalized root mean square error (RMSE_n) is used instead of RMSE,
since RMSE is scale-dependent, and RMSE_n is better suited for comparing the model performances (Sun, 2013).

The correlation coefficient (*r*) is a measurement degree of collinearity between observed and predicted data and the likeliness of
the outcomes to be predicted in the future.

$$\frac{\sum_{i=1}^{N}(y_i-\bar{y})(o_i-\bar{o})}{\sqrt{\sum_{i=1}^{N}(y_i-\bar{y})^2}\sqrt{\sum_{i=1}^{N}(o_i-\bar{o})^2}} \tag{6}$$

Next, the predictive skill of the model is evaluated with Nash-Sutcliff efficiency (*NSE*). NSE is defined as

$$NSE = 1 - \frac{\sum_{i=1}^{N}(y_i-\bar{y})^2}{\sum_{i=1}^{N}(o_i-\bar{o})^2} \tag{7}$$

Root mean square error (*RMSE*) provides fitness of the model (i.e., the deviation between the simulated and predicted values from
the actual values).

$$RMSE = \left(\frac{1}{N}\sum_{i=1}^{N}(y_i-o_i)^2\right)^{\frac{1}{2}} \tag{8}$$

$$RMSE_n = \frac{RMSE}{\sigma_{ob}} \tag{9}$$

Where the standard deviation of the observed values is $\sigma_{ob}$

The standard error of the predicted y is:

$$\sqrt{\frac{1}{(n-2)}\left[\sum(y_i-\bar{y})^2 - \frac{[\sum(o_i-\bar{o})(y_i-\bar{y})]^2}{\sum(o_i-\bar{o})^2}\right]} \tag{10}$$

Lastly, the percentage prediction error is defined as

Percentage prediction error $= \left[\frac{o_i-y_i}{o_i}\right]100 \tag{11}$

Where $y_i$ and $o_i$ are the predicted and observed values. $\bar{y}$, $\bar{o}$ denotes the mean of predicted and observed values, respectively.

It should be noted that the model performances are reported to be good if the resulting *NSE>0.65* and *RMSE_n <0.6* (Moriasi et al.,
2007). The models are analyzed for individual well scale to identify the spatial variability of the predictive performances.
Furthermore, basin and sub-basins scale median results are analyzed to observe the basin-wise performance and generalization
ability of the models. A flowchart of the detailed methodology followed for the ML modeling in the study has been shown in Figure
S3.

### 2.6 Limitations, assumptions, and uncertainty

The GWL data used in the study consists of a wide range of time periods, frequency, and temporal continuation. We used filtering
for removing the outliers and implemented imputation methods to fill the data gaps. The gridded datasets for precipitation,





temperature (maximum, mean, and minimum), and potential evapotranspiration are smoothed processed products obtained from
the actual observations, which have limitations in capturing the extreme climate events. This may affect the model development
when the climatic data are used as input. Moreover, the ML methods used in the study has some weaknesses regarding the low
generalizability of the methods, risk of overtraining. The majority of the observation wells (>87%) are in the unconfined aquifer
(CGWB, 2014). Due to the inaccessibility of groundwater level data, the Pakistan part of the Indus basin is not used in the study.
Abstraction and population could not be used as an explanatory variable in the machine learning modeling, due to paucity of spatio-
temporal data. Here, we have considered the population of the basin and sub-basins; and have not distinguished the effect of rural
vs. urban population in GWL change. The observation wells used in the study are not evenly distributed and occasionally clustered
towards the urban areas. Due to paucity, we had to work with a lower number of deep observation wells.

### 3 Results

### 3.1 Spatial patterns of model performances based on observation well scale analysis

We have developed ANN and SVM models and evaluated the relationship between the observed and simulated GWLs with r,
NSE, and $RMSE_n$ under significant p-value of ($p < 0.05$) for the test period (*2006 – 2010*) for each of the 2303 observation wells.
The results indicate that both the methodologies show good capability across the basin. However, relative weakening of spatial
correlations (Figure 2a, S5) have been observed in some of the specific regions in all the plots (Figure 2, S4, S5, S6), mostly in the
Indus basin; the western and eastern parts of the Ganges basin; eastern Meghna basin and parts of the Brahmaputra basin. The NSE
(Figure 2b, S6) and $RMSE_n$ (Figure 2c, S7) maps are mostly in agreement with the correlation map in terms of model performances.

ANN simulated GWL data matches fairly well with the observed GWLs in the testing period (2006 – 2015). High correlation is
observed in most parts of the basin with ~51%, ~58%, and ~48% of the observation wells having correlations greater than 0.6
($p<0.05$) for Model A, Model B, and Model C respectively (Table 1). The NSE results are promising, with wells having NSE
higher than 0.65 ($p<0.05$) for Model A, Model B, and Model C are ~62%, ~55%, ~40%, respectively (Table 1). Similarly, ~45%,
~57%, and ~36% of the wells have $RMSE_n$ less than 0.6 (Table 1).

Furthermore, SVM simulations of GWLs also show good results. The analysis based on SVM outcomes reveals that Model A,
Model B, and Model C observation wells with correlations greater than 0.6 (p<0.05) are ~63%, ~68%, ~51%, respectively (Table
1). Furthermore, wells with NSE higher than 0.65 are ~74%, ~83%, ~60% of the total number of observations well, respectively.
Similarly, ~65%, ~81%, and ~41% of the wells have $RMSE_n$ less than 0.6 (Table 1).

The number summaries (a) Correlation, (b) Nash-Sutcliff efficiency, and (c) Normalized Root mean square error between the
observed and simulated groundwater levels for all the ML models (ANN Model A, B, C; SVM Model A, B, C) have been presented
in Figure S8, S9, S10 respectively. Figure 3 shows the location wise median values of the correlation coefficient, Nash-Sutcliff
efficiency, and Normalized Root mean square error in boxplot format obtained from all the models. According to the Boxplots, an
improvement in model performances is observed from ANN to SVM, and SVM model B performs better than other ML models.

### 280 3.2 Basin-scale model performances

The performance matrices ( r, NSE, $RMSE_n$ ) with the median values for the entire study area for all the models (ANN Model A,
B, C; SVM Model A, B, C) are calculated. ANN models report very good correlation coefficient values (Table S6, S7) for training
and testing ($r_{med}>0.99, p<0.05$) stages. The NSE (Tables S8, S9) and $RMSE_n$ (Tables S10, S11) for both the training (NSE> 0.91,
$RMSE_n$ <0.30) and testing ($NSE_{med}$> 0.90, $RMSE_{n(med)}$ <0.30) data reflects good fit between the observed and simulated GWLs in
the IGBM basin scale. Furthermore, SVM models perform better than the ANN models for all three model types. In case of SVM,



we observe the following average statistics for training and testing stage: $r_{med} > 0.99$ ($p<0.05$), $NSE_{med} > 0.94$; and $RMSE_{n(med)} <$ 0.25.

Based on the calculated performance matrices from the observed and simulated median time series for both the training period (Table S6, S8, S10) and testing period (Table S7, S9, S11), the order of predictive performance (higher r, higher NSE, lower

$RMSE_n$) of the sub-basin are as follows: Ganges basin, Brahmaputra basin, Meghna basin, Indus basin. Comparative time series of the observed and predicted median groundwater levels for all the basin, sub-basin, and depth categories are shown in Figure 4.

### 3.3 Performance features linked to the depth of the observation wells

Our findings from observational well-scale analysis (Section 3.1) suggest that ~54% and ~66% of the shallow observation wells have correlations greater than 0.6 ($p<0.05$) for ANN and SVM, respectively (Table 1). Moreover, ~53% and ~78% of the shallow

observation wells show NSE >0.65 for ANN and SVM, respectively (Table 1). However, for deep observation wells, ~40% and ~60% have correlations greater than 0.6 ($p<0.05$), and ~48% and ~72% of the wells have NSE>0.65 for ANN and SVM respectively (Table 1). Similar to r and NSE, the number of wells decreases for $RMSE_n$ <0.6 from shallow to deep wells (Table 1). Furthermore, the basin-scale analysis (Section 3.2) suggests that for deep wells, a significant decline (41% and 19% decrease in r and NSE; 67 % increase in $RMSE_n$) in model efficiency, relative to the shallow wells for both ANN and SVM models in the test

period (Tables S7, S9. S11).

### 3.4 Model features linked to the population of the region

Due to lack of spatio-temporal data, the population could not be used as an explanatory variable in the machine learning modeling. Hence, to delineate the possible effect of the population on groundwater level in the IGBM basin, we have sub-divided the entire IGBM basin into three categories (Figure 5a) based on the average population count in a ~5 km grid using yearly data for 2000,

2005, 2010, 2015 (Fig S16). The categories are: Class 1 (population: <10,000), Class 2 (population: 10,000 – 20,000), and Class 3 (population: > 20,000).

Figure 5b demonstrates the percentage error of the observed and simulated GWLs from the models in each of the population classes. The results show that the low relative percentage error for Class 1; however, the error increases from Class 1 to Class 3. Other observations (on the comparative performance of ANN, SVM for Model A, B, C) are discussed in the previous sections.

## 4 Discussions

### 4.1 Relative importance of influencing factors in groundwater level change in the Indus-Ganges-Brahmaputra basin

We determined the relative influence/contribution of the explanatory variables as strong or weaker dominance over one another using dominance analysis, Here, the influence of precipitation, temperature, evapotranspiration, abstraction, and population on GWL is analyzed (Figure 6). Our analysis shows, groundwater withdrawals, and the population are the two most dominating

variables in the region. In general, the deep wells show more dominance by the abstraction than the shallower wells. The Indus and Meghna sub-basin are primarily dominated by groundwater abstraction. Groundwater abstraction in the Brahmaputra basin, shows a weaker dominance, while precipitation, evapotranspiration have relatively stronger dominance on GWLs. Temperature shows weaker dominance on GWLs, relative to the other variables except for the Indus and Meghna basins.

Significant groundwater abstraction is observed in most of the basin, especially in the Indus, Meghna basin, and Bengal part of the

Ganges basin, where yearly groundwater withdrawals can be as high as 900 million cubic meters or more (Figure 1c). The uncontrolled irrigation practices (Barik et al., 2016; Bhanja et al., 2017) lead to the over-exploitation of the aquifers in the IGBM,





which is reflected in the deepening of GWL in north-west India, south-east Bangladesh, western Ganges basin and Bangal part of Ganges basin in the east (Figure 1b).

**4.2 Assessment of spatial variability of machine learning model performances**

The IGBM alluvial aquifer system exhibits significant spatial variation in hydraulic properties, groundwater recharge, storage, and abstraction (MacDonald et al., 2016; Bonsor et al., 2017; Bhanja et al., 2018). Therefore, it is crucial to analyze each observation wells to show the spatial variability in model performances. Our results based on individual well-scale (Table 1, Figure 2) and basin-scale analysis (Table S7, S9, S11) reveal that the ANN and SVM models have limitations in areas with higher groundwater abstraction (Figure 1c). Hence, poor results are observed in the Indus and Meghna basin (Figure 2). However, in the Brahmaputra

basin, where precipitation is the main controlling factor, and groundwater abstraction is relatively low, it still demonstrates relatively moderate model performances. It should be noted, that the results of the well scale analysis (Table 1) are in agreement with the basin-scale analysis (Table S7, S9, S11). It is also worth mentioning that after the inclusion of meteorological variables in the system along with the GWLs (Model B) as input variables, the performance of the models improves significantly. The variability of model performances is strongly linked with the aquifer response to groundwater abstraction and recharge processes.

**4.3 Depth-wise assessments of model performances**

The number of operational shallow irrigational wells show a decrease in the post-2005 time period and a continuous increase in deep irrigational well in post-2000 (Figure S17). The change in the irrigational pattern is adapted mainly to mine cleaner deeper groundwater due to the continued lowering of GWLs and contaminated shallow water (Famiglietti, 2014; Fendorf et al., 2010) in the basin. Furthermore, the output of a deep irrigational well is approximately 15 times higher than a shallow well (Minor irrigation,

2017). This leads to a rapid decline in deep groundwater with respect to the shallow groundwater. Hence, the deterioration of model performances with increasing intake depth in the subsurface is linked to the dominance of groundwater abstraction from the deeper depths of the aquifer (Figure 6), which is significant for Indus, Meghna, and Brahmaputra basin. However, this observation for the Brahmaputra basin is less reliable for the use of a comparatively lower number of deep wells in the basin.

**4.4 Effect of population on groundwater level modeling**

Our results obtained from the dominance analysis reveal that the population of the region is an important predictor variable for GWL in the IGBM basin. Additional analysis has been implemented to comprehend the predictive performance of all the ANN and SVM model types (Model A, Model B, Model C) to the population in the region. We find a proportional relationship between GWL decline and population increase from the dominance analysis (Figure 6). This could be a possible explanation for poor model performances in the areas of the dense population (Figure 5a,b). Please note that water use may not be strongly correlated to

population, especially in rural areas, where pumping for irrigation is not necessarily linked to population.

**4.5 Groundwater level modeling using only climatic variables**

An important finding of the study is that the models are also capable of simulating GWLs with only using the meteorological variables (Model C), in certain areas where groundwater abstraction is not the major controlling factor in groundwater level change. The performance of the models are less accurate than the models trained using GWL as input. The results obtained from the ML

models and the dominance analysis show that in exploited regions, where the human effects obscure the natural dynamics of the hydrologic system, the models using only climate variables do not perform efficiently.

**4.6 Sensitivity analysis**





In order to investigate the error associated with the number of years of input and output in the ML methods, we used the best performing model (Model B here) for the shallow wells (as the spatial variability is better for shallow wells with a good number of measurement location availability) in the IGBM basin. The number of input and output data configurations are shown in Table S5, and the results are shown in Fig. S4. Our analyses show the best model performance on using maximum years of input data (Configuration - 5), based on the correlation coefficient, NSE, and standard error between observed and simulated GWLs. These analyses enable users to use their own time-series configuration with a pre-determined prediction error. The users could perform the prediction analysis after looking at the number of years of prediction without compromising the output quality within the study region or at a region with a similar hydroclimatic setting.

## 5 Conclusions

In this study, we compared the observed and machine learning simulated GWL time series for in-situ observation wells (*n=2303*, with intake depth information) using past GWLs and/or climatic variables across the transboundary aquifers of India and Bangladesh. Our results indicate a strong relationship between the observed and simulated GWLs in most of the basin. We find improvement in model performances when a greater number of years are used as input data. However, in the intensively irrigated agricultural regions, a poor relationship is observed, suggesting human influence affecting the natural dynamics of the groundwater system. The results from the dominance analysis suggest that groundwater abstraction and population are the most important predictor variables in the region. The ML-based models are also in agreement with dominance analysis in the sense that the model performance reduced in high abstraction and high populated areas of IGBM. Furthermore, we noted that, the in areas with low groundwater abstraction climate variables could be effectively used for GWL simulation using ML. Furthermore, model efficiency improves when climatic variables are included as input variables in addition to past GWLs into the system. Therefore, the general consensus based on the noticeable spatial differences in model performances suggests the model's limitations to provide good results in areas dominated by anthropogenic factors. The measurement and predictive analysis of the groundwater system is an important task to quantify the present and future groundwater resources. With the exponentially growing population, sustainable development depends on the efficient national and regional level adaptation policies guiding the agricultural priorities in risk reduction. The findings of the study may help in understanding the GWL dynamics in the aquifers for building regional-scale prediction models for sustainable governance of groundwater resources in other parts of the world.

**Code and Data Availability:**

Code is available on request. This study uses open-source data sets. Groundwater level and abstraction data were obtained from the Central Ground Water Board (http://cgwb.gov.in/index.html) of Government of India and Bangladesh Water Development Board (https://www.bwdb.gov.bd/) of Government of Bangladesh. Climate data was available from the India Meteorological Department (IMD) (http://www.imd.gov.in/Welcome%20To%20IMD/Welcome.php) and Climatic Research Unit (CRU TS v-4.01) (https://crudata.uea.ac.uk/cru/data/hrg/). Population data obtained from Palisades NY: NASA Socioeconomic Data and Applications Center (SEDAC) (https://sedac.ciesin.columbia.edu/data/set/gpw-v4-population-count-rev11).

**Author contribution statement**

P.M. and A.M. designed the study. P.M. performed background analyses and simulations. P.M. did the study under the supervision of A.M. and advice from S.N.B. Data management and processing were performed by PM with inputs from S.N.B.; R.K.R., D.S, and A.Z. helped with the data retrieval. P.M. wrote the manuscript with inputs from A.M., S.N.B., D.S., R.K.R., S.S., A.Z.

**Declaration of Competing Interest**

The authors declare that they have no conflict of interest.





**Acknowledgments**

The authors acknowledge Ministry of Jal Shakti, Department of Water Resources, River Development and Ganga Rejuvenation of Government of India and Bangladesh Water Development Board of Government of Bangladesh, India Meteorological Department

(IMD), Climatic Research Unit (CRU), AQUASTAT (http://www.fao.org) for data support. The authors are thankful to Palash Debnath, Srimanti DattaGupta, Kousik Das at IIT Kharagpur. The authors acknowledge the use of ArcGIS software (version 10.2.1), Origin software (version 2015), Matlab 2015a, R statistical software, and Ferret (http://ferret.pmel.noaa.gov/Ferret/. Pacific Marine Environmental Laboratory (NOAA).

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

**Table 1.** Percentage of observation wells having correlation coefficient (r) greater than 0.6, Nash-Sutcliff efficiency (NSE) greater than 0.65, Normalized Root mean square error ($RMSE_n$) is less than 0.6 based on the well-scale analysis.

| | ANN | | | SVM | | |
|---|---|---|---|---|---|---|
| | Model A | Model B | Model C | Model A | Model B | Model C |
| **Correlation coefficient greater (r) > 0.6** | | | | | | |
| ALL | 51.3 | 57.5 | 48.1 | 63.2 | 67.7 | 51.4 |
| SH | 49.6 | 58.7 | 52.7 | 72.6 | 75.7 | 51.5 |
| DP | 34 | 46.6 | 38.1 | 62.2 | 66.8 | 51.3 |
| **Nash-Sutcliff efficiency (NSE) > 0.65** | | | | | | |
| ALL | 61.8 | 55.4 | 40 | 73.9 | 82.9 | 60.3 |
| SH | 50.6 | 65.4 | 42.1 | 78.9 | 85.6 | 68.6 |
| DP | 49.4 | 54.4 | 39.8 | 73.4 | 82.6 | 59.5 |
| **Normalized Root mean square error ($RMSE_n$) < 0.6** | | | | | | |
| ALL | 45.4 | 56.7 | 36.4 | 64.7 | 81.4 | 40.5 |
| SH | 46.3 | 57.8 | 37.4 | 65.8 | 82.8 | 41.7 |
| DP | 35.4 | 44.3 | 26 | 53.8 | 65.9 | 28.6 |










**Figures:**

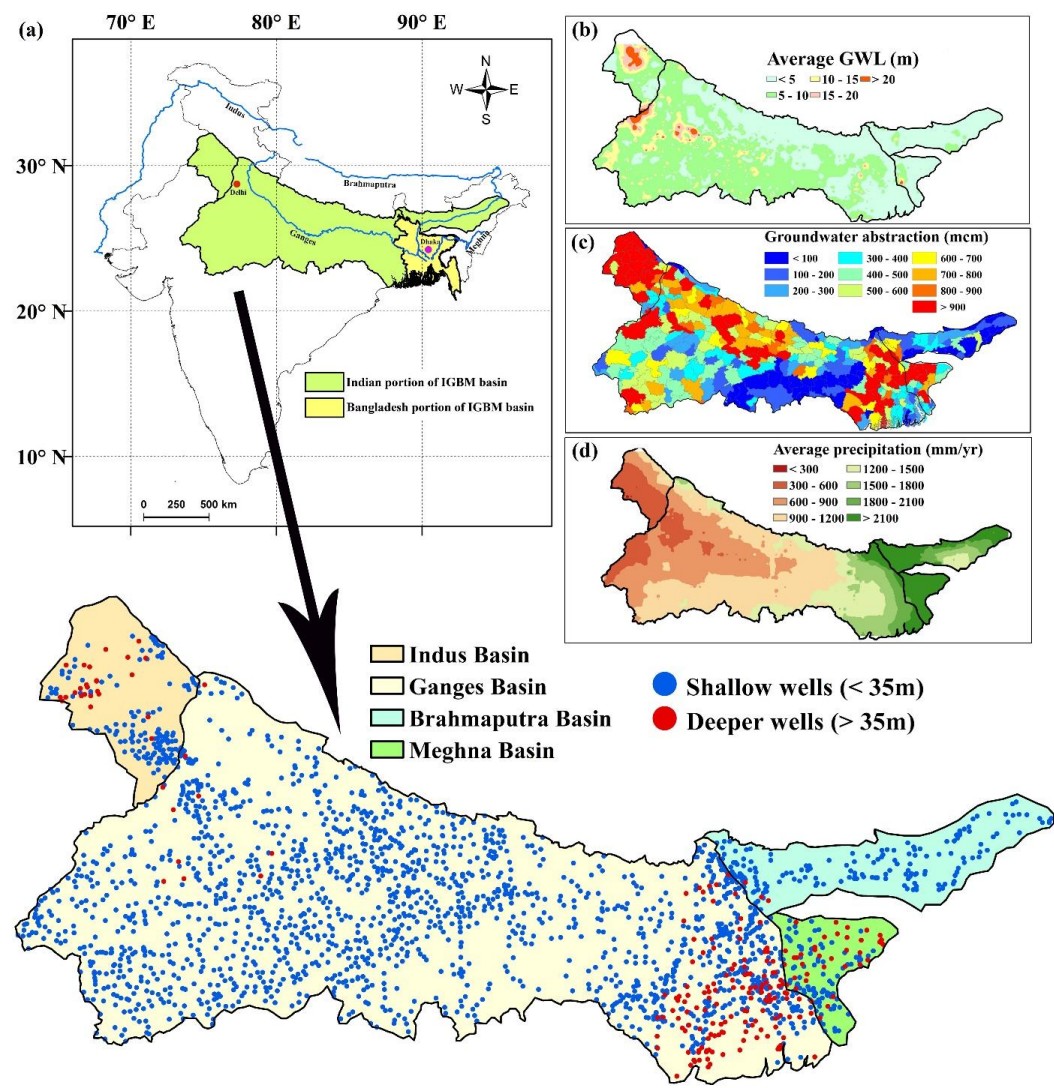


Figure 1. (a) Map of the study area, location of the IGBM basin and sub-basins with 2303 groundwater observation wells (Shallow observation wells: intake depth <35m, n= 2080 and deep observation wells; intake depth >35 m; n = 223); (b) Average groundwater level map between 1985 to 2015; (c) District level groundwater abstraction in million cubic meters (mcm) for 2013 in the IGMB basin; (d) Long term mean annual precipitation (mm/year) distribution in the IGBM basin from 1985 to 2015.


**Figure 2. (a)** Correlation coefficient, **(b)** Nash-Sutcliff efficiency, and **(c)** Normalized Root mean square error map between observed and simulated groundwater levels. These maps are generated with the location wise median values of the correlation coefficient, Nash-Sutcliff efficiency, and Normalized Root mean square error obtained from all the models (ANN Model A, B, C; SVM Model A, B, C) for the testing period.




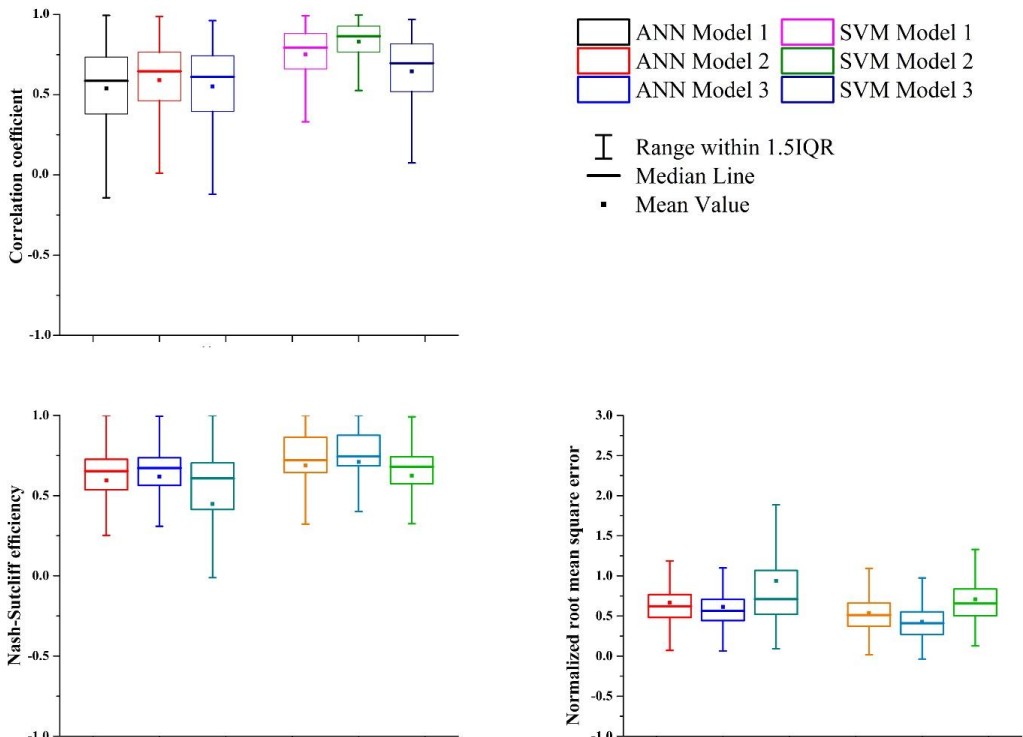

**Figure 3. Boxplots for (a)** Correlation coefficient, **(b)** Nash-Sutcliff efficiency, and **(c)** Normalized Root mean square error between observed and simulated groundwater levels. These boxplots are generated with the location wise median values of the correlation coefficient, Nash-Sutcliff efficiency, and Normalized Root mean square error obtained from all the models (ANN Model A, B, C; SVM Model A, B, C) for the testing period.




**Figure 4. Comparative time series of the observed and simulated median groundwater levels for all the basin, sub-basin, and depth categories using ANN and SVM.**

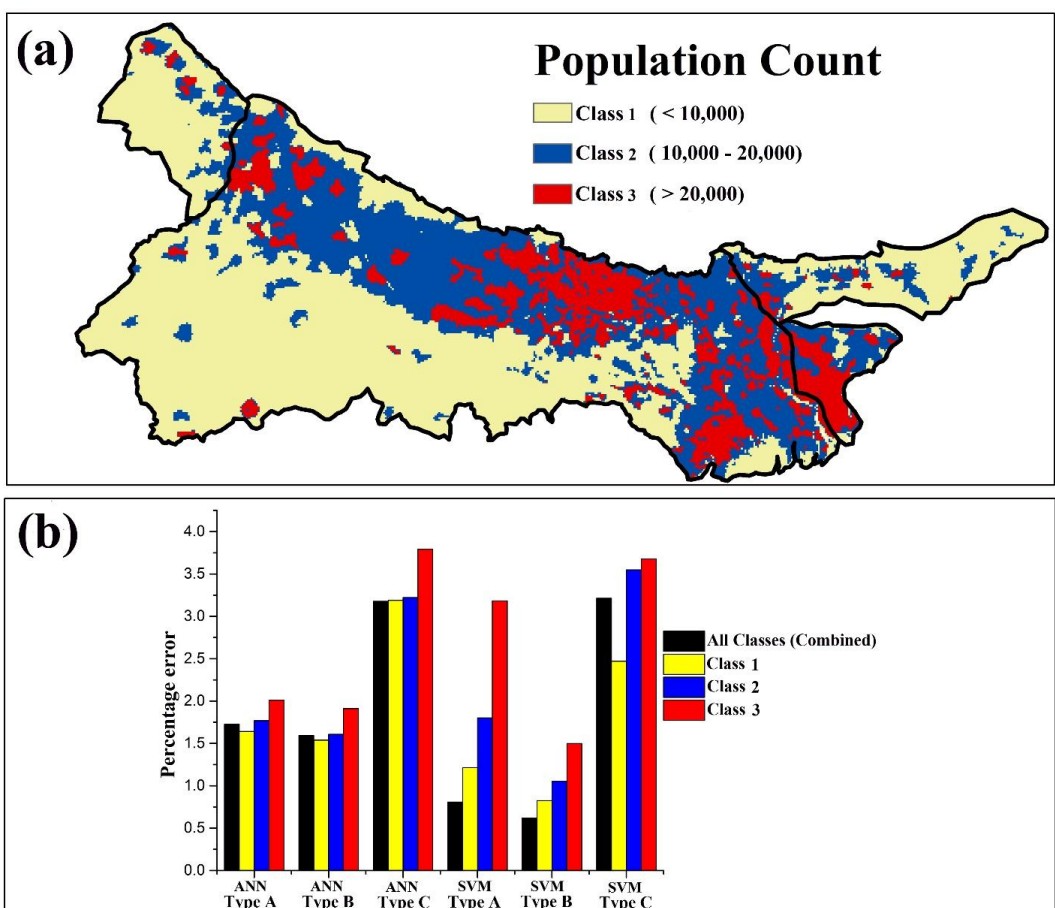

Figure 5. (a) Population classes based on the population count in ~5km grids; (b) percentage error of ANN and SVM models
       for different population classes.

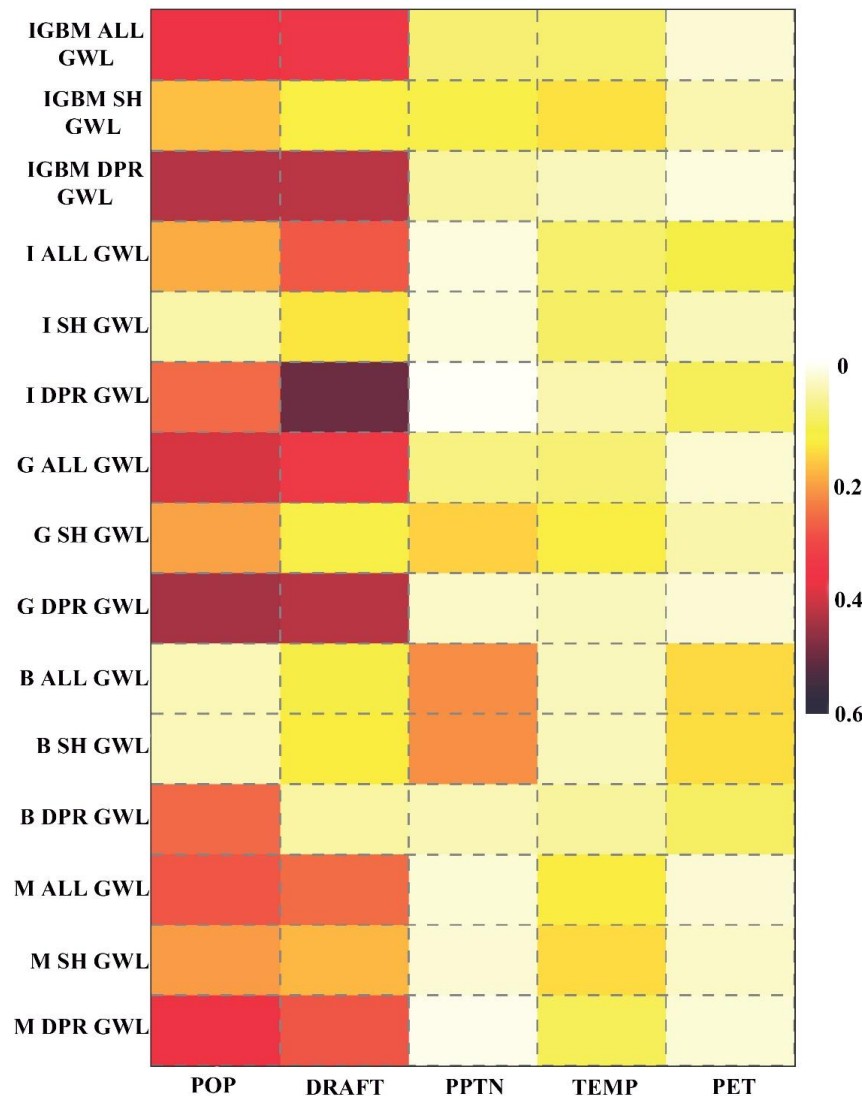

**Figure 6. The relative contribution of the predictor variables on groundwater level variation, determined by the dominance analysis.**