# Peer review of "Importance of spatial and depth-dependent drivers in groundwater level modeling through machine learning"

_Hydrology and Earth System Sciences, 2020_

## Short Comment (SC1) · 4 Jun 2020

The authors have treated an interesting topic dealing with groundwater in a large transboundary aquifer between India and Bangladesh for the purpose of highlighting the influence of various triggers; natural and anthropogenic that act and harm this groundwater. The investigation is carried out by the use of machine learning methods (support vector machines and artificial neural network The application of this kind of modeling constitutes a novelty for groundwater in the studied basin. The title is appropriate for the content of the paper, however, it will be better if they add an indication about the study area. The abstract summarizes the main information of the paper and highlights

the main finding. The paper is well written and balanced. However, the article contains imperfections such as: • The authors repeatedly cited Figures S1 up to S17 (line 263) and Tables S1 to S11 (example line 282, 283) but in the list of figures and tables below those illustrations are missing. • Line 47 they wrote: south-east India (Bengal basin), maybe they rather say North-east India. • Line 49, do ‶Summer" and ‶winter" correspond to ‶Rabi" and ‶Kharif" respectively. • Line 54, could you give value to the population? • Line 166, authors should add a reference. • In all the manuscript, when a cardinal point is preceded by "the" the first letter should be written in capital letter. • Line 349, it's better to change ‶Please" Âż by ‶we". For references : • In the reference list, the citations should be written in the alphabetic order. • For some references with the same first author, they should be classified ..YEARa, YEARb. . ., example ‶Bhandri et al, 2019" • There is a disagreement between some references in the text and in the list, for example, Youn et al, 2016, and in the list 2011. BADC, 2017 in the text and 2014 on the list. • The reference SEDAC, 2018 is missing n the list. • The reference Bhanja et al, 2017 is missing in the list.

In conclusion, I recommend that this paper will be accepted after minor revisions.

Please also note the supplement to this comment:
https://www.hydrol-earth-syst-sci-discuss.net/hess-2020-208/hess-2020-208-SC1-supplement.pdf

---

## Referee Comment (RC1) · Anonymous Referee #1 · 23 Jun 2020

This study investigated the spatial patterns of the performance of machine learning method for the groundwater level modeling using a large number of observations. the topic of this manuscript is interesting, considering the importance of groundwater resources management in the south Asia. I think this manuscript can be accepted after major revisions. The comments are given below:

Major comments

1. The SVM performs better than ANN model, the reason behind this result should be explained.

2. Figure 4. For Bhahmaputra(DP), it seems that all the models show large errors in

the testing period. The model may loss stability in this region for deep depth modeling. This needs explanations.

3. Section 2.1. The aquifer is heterogeneous. The spatial variations in permeability and other hydrogeology conditions, such as the character of the rocks, the depth of the aquifer, etc, need to be described here.

4. Line 145 "The missing values in the GWL time series data were filled using Multiple imputation". It needs to be clarified that how many wells have missing data. Did filling missing data influence the result? A comparison of modeling results at wells with missing data and other wells without missing data is needed.

5. I suggest the author to add a discussion with previous similar studies to illustrate the differences between this study and other studies.

Minor comments

1. Figure S14. There are errors for the label of y-axis."SVM A" should not be followed by "ANN B" and "ANN C"

2. Figure S3. I suggest the flowchart can be moved to the main text.

3. The manuscript analyzed the influence of population and groundwater withdrawal. These two variables may be related, this need to be clarified, and what are the purpose for groundwater abstraction?

---

## Short Comment (SC2) · 23 Jun 2020

The authors have investigated the relative influence of major drivers in groundwater level change and linked them with the performance of machine learning-based predictive models, in a very important transboundary system. The study illustrates the advantages and limitations of machine learning-based modeling in a very heterogeneous regime. Due to this specific study area, this study is particularly important. The spatial and depth-dependent variability in model performance using GWL data is novel. The depth component of the study is particularly impressive, and probably first of its kind. In my view, the manuscript should be accepted with minor revision.

[Figure]

The manuscript is well written, well-segmented and concise. However, there are some typos that should be corrected.

I have a few suggestions.

1) If possible please add few lines on major drivers in the introduction section, importantly for the abstracted part of the aquifer.

2) It would be better if the flow chart is moved into the main article from the supplementary section.

3) The Geology and hydrology of the study area could be expanded a little more.

4) In Figure 6 please mention the full form of the abbreviation used, at least in the figure captions.

5) In the ANN, SVM table (Table 1) the author should explain in short Model A, B, C.

---

## Referee Comment (RC2) · Anonymous Referee #2 · 26 Sep 2020

Comments on Manuscript Hess 2020-208-manuscript 'Importance of spatial and depth-dependent drivers in groundwater level modeling through machine learning'

Groundwater is an important source of water, in particular for the transboundary areas of IGBM Rivers. This study used a liner regression approach based on dominance analysis and machine learning methods to identify the spatial and depth-wise drivers based on a large network of ground-based observations. Some interesting conclusions are found by the authors, including e.g. the groundwater level change is primarily influenced by abstraction and population in most of the IGBM; the machine learning methods can well simulate the groundwater level and the performance decreases from

shallow to deep observation wells. The conclusions can be useful for groundwater management in the IGBM areas. However, the quality of this manuscript is not good enough for publication in HESS. The detailed comments are shown as the following.

Detailed comments:

1) Machine learning methods are popular over the years. The authors gave an introduction to machine learning methods used in GWL. I expect that more prevailing methods should be mentioned in the introduction instead of ANN and SVR. I also expect a comparison of these prevailing methods. 2) From the manuscript, it is difficult to see the originality of this study. For me, the only originality might be the use of a large network of monitoring wells to identify the spatial and depth-wise drivers. 3) Line 120: Although a large network of monitoring wells was used, the time resolution is rather coarse. Also can the authors show us the time series of monitored water levels? 4) For the dominance analysis, the independent variables seem dependent, such as groundwater withdrawals and population, temperature and potential evapotranspiration. Will this affect the results of dominance analysis? 5) Section 2.5: I am curious why the authors used two somewhat old-fashioned models including ANN and SVM. It is very easy to over-train these two types of models. I suggest the authors to use other models including LSTM. 6) Line 251: replace 'has' with 'have' 7) If the ML methods used in the study have some weakness regarding the low generalizability of the methods, risk of overtraining, why did the authors choose other machine learning methods? 8) Line 268: it seems to me that only half of the observation wells having correlations greater than 0.6 is not much. 9) Line 328: it is expected that the ANN and SVM models have limitations in areas with higher groundwater abstraction. 10) Figure 4: why large deviations in Indus? 11) Figure 6: how were the relative contributions calculated? Based on coefficient of determination?

---

## Author Comment (AC1) · 22 Oct 2020

*"Importance of spatial and depth-dependent drivers in groundwater level modeling through machine learning"* by Pragnaditya Malakar, Abhijit Mukherjee, Soumendra N. Bhanja, Dipankar Saha, Ranjan Kumar Ray, Sudeshna Sarkar, Anwar Zahid

**Reviewer #1:** This study investigated the spatial patterns of the performance of machine learning method for the groundwater level modeling using a large number of observations. the topic of this manuscript is interesting, considering the importance of groundwater resources management in the south Asia. I think this manuscript can be accepted after major revisions. The comments are given below:

Reply: We would like to thank the reviewer for his/her appreciation. We have addressed the reviewer's comments and done a complete major revision of the manuscript. Doing so, we have added details on the method, results, and discussion, rewritten several sections, and added a few new figures, which we believe have greatly improved the manuscript.

**Highlights of the revision:**

We have

  a) Explained the potential rationale behind SVM's better performance than ANN
  b) Described the hydrogeological conditions and aquifer characteristics and two maps added in this regard
  c) Added two figures and a table showing data availability of the observation wells and also included analyses to show the influence of missing data on model performances
  d) Including a detailed discussion of the previous studies, differences with our study, and originality of the present study
  e) Moved the flowchart to the main text

**Rev 1. Major Comment 1:** The SVM performs better than ANN model, the reason behind this result should be explained.

Reply: We would like to thank the reviewer for the comment. In this study, our findings suggest that SVM performs better than ANN, which is in agreement with other studies *(Yoon et al., 2011; Yoon et al., 2016; Mukherjee et al., 2018, Bhanja et al., 2019)* on groundwater level prediction or

simulation. The exact reason for the improved model performance is difficult to explain since the models have different structures. One of the potential reasons could be SVM's better robustness than ANN for its application on groundwater level time-series data. Following the reviewer's comment, we have added a paragraph in the Discussion section.

We added,

*"In this context, we used similar input data in ANN and SVM to compare the performance of these computational methods. Our results are in general agreement with the previous studies (Yoon et al., 2011; Yoon et al., 2016; Mukherjee et al., 2018, Bhanja et al., 2019) on SVM performing better than ANN in groundwater level time series simulation and prediction. The exact rationale behind the SVM model performance improvement is difficult to explain since the models have different structures. A potential reason could be linked to SVM's ability of converging on global minimum and allowing a better tolerance to the noise (based on the inherent pattern associated with the datasets). Thus SVM may have certain benefits over ANN regarding the robustness and convergence (Burges et al., 1998; Karamizadeh et al., 2014). Thus, the results further highlight the importance of developing multiple methods for groundwater level modeling using machine learning. The comparison may indicate the best way forward, which is one of the motivations of this study."*

**References**

Burges, C. J. C.: A tutorial on support vector machines for pattern recognition, Data Min. Knowl. Discov., 2(2), 121–167, doi:10.1023/A:1009715923555, 1998.

Karamizadeh, S., Abdullah, S. M., Halimi, M., Shayan, J. and Rajabi, M. J.: Advantage and drawback of support vector machine functionality, I4CT 2014 - 1st Int. Conf. Comput. Commun. Control Technol. Proc., 63–65, doi:10.1109/I4CT.2014.6914146, 2014.

Mukherjee, A. and Ramachandran, P.: Prediction of GWL with the help of GRACE TWS for unevenly spaced time series data in India : Analysis of comparative performances of SVR, ANN and LRM, J. Hydrol., 558(October 2008), 647–658, doi:10.1016/j.jhydrol.2018.02.005, 2018.

Yoon, H., Jun, S. C., Hyun, Y., Bae, G. O. and Lee, K. K.: A comparative study of artificial neural networks and support vector machines for predicting groundwater levels in a coastal aquifer, J. Hydrol., 396(1–2), 128–138, doi:10.1016/j.jhydrol.2010.11.002, 2011.

Yoon, H., Hyun, Y., Ha, K., Lee, K. K. and Kim, G. B.: A method to improve the stability and accuracy of ANN- and SVM-based time series models for long-term groundwater level predictions, Comput. Geosci., 90, 144–155, doi:10.1016/j.cageo.2016.03.002, 2016.

**Rev 1. Major Comment 2:** Figure 4. For Bhahmaputra (DP), it seems that all the models show large errors in the testing period. The model may loss stability in this region for deep depth modeling. This needs explanations.

Reply: We thank the reviewer for the comment. Figure 4 demonstrates the comparative time series of the observed and simulated median groundwater levels for all the basin, sub-basin, and depth categories using ANN and SVM. As pointed by the reviewer, it is evident that Brahmaputra (DP) shows a large error. Please note that there are only three observation wells in the Brahmaputra (DP) category. Thus from a statistical perspective, the median observed and simulated median groundwater levels are less reliable.

We have mentioned this in the text.

"*However, the DP observations for the Brahmaputra basin is less reliable for the use of a comparatively lower number of deep wells (Table S4) in the basin.*"

**Rev 1. Major Comment 3:** Section 2.1. The aquifer is heterogeneous. The spatial variations in permeability and other hydrogeology conditions, such as the character of the rocks, the depth of the aquifer, etc, need to be described here.

Reply: We thank the reviewer for his/her comment. The IGBM basin exhibits a wide range of permeability, transmissibility, hydraulic conductivity, and aquifer depth. The diverse depositional settings and environment of Pleistocene to Holocene sediments resulted in variable aquifer properties across the basin. Following the reviewer's suggestions, we have added a brief description of the hydrogeological conditions and aquifer characteristics of the IGBM.

Furthermore, we also added two figures showing the aquifer type, horizontal hydraulic conductivity, transmissivity, and specific yield of India and Bangladesh.

We added,

"*The sediment (both recent Plio‑Pleistocene to Holocene alluvium and older Miocene rocks) thickness of IGBM is up to 2 km (Singh et al., 1996). However, the effective thickness of the aquifer in most of the IGBM is generally the top 200 m. Notably, in the Bengal basin area in the eastern part of the Ganges basin and the Indus basin area, the effective aquifer thickness could be greater than 300 m (Mukherjee et al., 2007; Macdonald et al., 2015; Bonsor et al., 2017). The diverse depositional setting and environment of Pleistocene to Holocene sediments resulted in variable aquifer properties across the basin (Bonsor et al., 2017). A distinct systematic reduction in permeability is found away from the mountain and towards the coast in most of the IGBM; however, the distribution is more complex for the Ganges basin (Macdonald et al., 2015). The transmissivity within the upper and middle Ganges basin and most of the Brahmaputra basin ranges from several 100 $m^2 day^{-1}$ to more than 5000 $m^2 day^{-1}$ (Bonsor et al., 2017), which is representative of permeability values of 5 – 100 m/d (CGWB 2010). However, in the Indus basin, the permeability values of <10 $m˙day^{-1}$ to >60m $m^2 day^{-1}$ is reported. Figure S1 show the aquifer type, horizontal hydraulic conductivity, and transmissivity of India and Bangladesh (Bhanja et al., 2017a, 2019a). The specific yield in the unconsolidated sedimentary (high hydraulic conductivity) aquifer part of the IGBM ranges from 0.06 to 0.20 (mean 0.013). However, the specific yield values up to 0.08 are reported in the consolidated sedimentary (medium hydraulic conductivity) part of the basin (Bhanja et al., 2016). A specific yield map for India and Bangladesh is shown in Figure S2.*"

[Figure]

*Figure S1. Different aquifer types, horizontal hydraulic conductivity (mday$^{-1}$) and transmissivity (m$^2$day$^{-1}$) for India and Bangladesh (modified from Bhanja et al., 2019a).*

[Figure]

*Figure. S2. Specific yield map for India and Bangladesh (modified from Bhanja et al., 2016).*

**Reference**

Bonsor, H. C., MacDonald, A. M., Ahmed, K. M., Burgess, W. G., Basharat, M., Calow, R. C., Dixit, A., Foster, S. S. D., Gopal, K., Lapworth, D. J., Moench, M., Mukherjee, A., Rao, M. S., Shamsudduha, M., Smith, L., Taylor, R. G., Tucker, J., van Steenbergen, F., Yadav, S. K. and Zahid, A.: Hydrogeological typologies of the Indo-Gangetic basin alluvial aquifer, South Asia, Hydrogeol. J., 25(5), 1377–1406, doi:10.1007/s10040-017-1550-z, 2017.

Bhanja, S. N., Mukherjee, A., Saha, D., Velicogna, I. and Famiglietti, J. S.: Validation of GRACE based groundwater storage anomaly using in-situ groundwater level measurements in India, J. Hydrol., 543, 729–738, doi:10.1016/j.jhydrol.2016.10.042, 2016.

Bhanja, S. N., Mukherjee, A., Rodell, M., Wada, Y., Chattopadhyay, S., Velicogna, I., Pangaluru, K. and Famiglietti, J. S.: Groundwater rejuvenation in parts of India influenced by water-policy change implementation, Sci. Rep., 7(1), 7453, doi:10.1038/s41598-017-07058-2, 2017a.

Bhanja, S. N., Mukherjee, A., Rangarajan, R., Scanlon, B. R., Malakar, P. and Verma, S.: Long-term groundwater recharge rates across India by in situ measurements, Hydrol. Earth Syst. Sci., 23(2), 711–722, doi:10.5194/hess-23-711-2019, 2019a.

Central Ground Water Board: Groundwater quality in shallow aquifers of India, Govt. of India, Ministry of Water Resources, Faridabad, 76 pp., 2010

MacDonald, A.M., Bonsor, H. C., Taylor, R., Shamsudduha, M., Burgess, W. G., Ahmed, K. M., Mukherjee, A., Zahid, A., Lapworth, D., Gopal, K., Rao, M. S., Moench, M., Bricker, S. H., Yadav, S. K., Satyal, Y., Smith, L., Dixit, A., Bell, R., van Steenbergen, F., Basharat, M., Gohar, M. S., Tucker, J., Calow, R. . C. and Maurice, L.: Groundwater resources in the Indo-Gangetic Basin: resilience to climate change and abstraction, 2015.

Mukherjee, A., Fryar, A. E. and Howell, P. D.: Regional hydrostratigraphy and groundwater flow modeling in the arsenic-affected areas of the western Bengal basin, West Bengal, India, Hydrogeol. J., 15(7), 1397–1418, doi:10.1007/s10040-007-0208-7, 2007.

Singh, I.B.: Geological evolution of Ganga Plain: an overview. J Palaeontol Soc India 41:99–137, 1996

**Rev 1. Major Comment 4:** Line 145 "The missing values in the GWL time series data were filled using Multiple imputation". It needs to be clarified that how many wells have missing data. Did filling missing data influence the result? A comparison of modeling results at wells with missing data and other wells without missing data is needed.

Reply: We thank the reviewer for his/her comment.

Following the reviewer's comment, we have added a spatial figure showing the data availability of each observation wells. We have also added a histogram showing the distribution of the number of observation wells (x axis) with groundwater level data records (y axis).

We added in the text,

*"The location wise GWL data availability is shown in Figure S4."*

[Figure]

*"Figure S4. Location-wise groundwater level data availability (%)."*

We also modified,

*"The usable number of observation wells (Figure S5) was reduced significantly (from n=13465 to usable n=2303), following the application of these filters and data processing."*

[Figure]

*"Figure S5. Summary of observation well data included in the study."*

To show the influence of missing data, we divided the observation wells into five categories (i.e., 75-80%, 80-85%, 85-90%, 90-95%, and 95-100% data availability) based on the data availability. We also computed the model performance for these five categories of data. We have included these results and discussed the finding in the revised manuscript. In general, the result suggests slightly better performance in the observation wells with higher data availability. However, no major increase or decrease in correlation coefficient, Nash-Sutcliff efficiency, and standard error are observed.

*"Groundwater level dataset contains few observation wells with no missing data (i.e., 100% data availability). Thus, to show the influence of missing data on groundwater level simulation results, we divided the observation wells according to highest to lowest data availability (Figure S5; Table S5). Similar to the sensitivity analysis, we used the best performing model (Model B) for the shallow wells (as the spatial variability is better for shallow wells with a good number of measurement location availability) in the IGBM basin. In general, the result (Figure S18) suggests slightly better performance for the observation wells with higher data availability. However, no major increase or decrease in correlation coefficient, Nash-Sutcliff efficiency, and standard error are observed (Figure S18)."*

**Table S5**. *Classes of observation wells adapted to observe the effect of missing data on model performance.*

| Data availability (%) | ANN | SVM | Number of wells |
|---|---|---|---|
| 100 - 95 | ANN_data availability=100%-95% | SVM_data availability=100%-95% | 275 |
| 95 - 90 | ANN_data availability=95%-90% | SVM_data availability=95%-90% | 430 |
| 90 - 85 | ANN_data availability=90%-85% | SVM_data availability=90%-85% | 423 |
| 85 - 80 | ANN_data availability=85%-80% | SVM_data availability=85%-80% | 442 |
| 80 - 75 | ANN_data availability=80%-75% | SVM_data availability=80%-75% | 510 |

[Figure]

*"Figure S18. Comparison of model performances with data availability."*

**Rev 1. Major Comment 5:** I suggest the author to add a discussion with previous similar studies to illustrate the differences between this study and other studies.

Reply: We thank the reviewer for the comment. In recent times, machine learning-based methods have been widely used to simulate and predict groundwater levels across the world. Most of these studies used methods like autoregressive integrated moving average (ARIMA), Artificial Neural Network (ANN), hybrid-ANN, Adaptive neuro-fuzzy inference system (ANFIS), genetic programming (GP), Support Vector Machine (SVM) and nonlinear auto-regressive exogenous model-based (NARX), long-short term memory (LSTM) model few others using a wide range of frequency and temporal data on past GWLs, satellite observations derived groundwater storage (GWS), Normalized difference vegetation index (NDVI)), meteorological variables, river discharge, variables of groundwater use, few dummy variables to simulate and/or predict GWLs. However, most studies (including studies on India and Bangladesh) are mainly small-scale studies, and due to the small number of observation wells, they are unable to characterize the spatial variability in model performances extensively. Moreover, the temporal extent of the studies on India and Bangladesh is often short. Hence the predictions are based on the short-term trends of dependent variables and do not consider the long-term variability. Furthermore, to our knowledge, none of the studies have considered the spatial and depth-wise performance variability of machine learning models in predicting GWL. The originality of this study lies in addressing some critical aspects which were not included in the previous studies. Firstly, to understand the spatial variability in machine learning-based model performances, we have considered a large network of monitoring wells (n = 2303) from 1985 to 2015 to simulate GWLs in the IGBM. Secondly, considering the variable depth-wise patterns of groundwater abstraction, we showed the significance of well depth (intake depth of the observation wells) information in GWL modeling using machine learning. Thirdly, we used meteorological variables exclusively to simulate in-situ GWLs. Fourthly, based on dominance analysis and outputs from the machine learning models, we investigated the most influential basin specific predictor(s) (both natural and human-induced) in GWL modeling. Following the reviewer's suggestion, we have modified the text, including the findings of the previous studies, and added new text to show the differences and originality of the present study.

*"Over the years, the simplistic approach and acceptable results of the machine learning (ML) methods are preferred when the underlying physical system is not well understood. Sometimes, decoding the physical system becomes much more complex due to interactions and feedbacks between multiple processes. GWL modeling based on ML has the unique ability to find the likely relationships between GWL and controlling hydro-climatic-anthropogenic variables without constructing knowledge-driven conceptual or physically-based models. Therefore, researchers have studied the performance of ML methods for GWL modeling in India and Bangladesh (Nayak et al., 2006; Nury et al., 2017; Malakar et al., 2018; Mukherjee and Ramachandran, 2018; Bhanja et al., 2019b; Sun et al., 2019; Yadav et al., 2020; and the references therein) and other parts of the world (Coulibaly et al., 2001; Feng et al., 2008; Sun, 2013; Nourani and Mousavi, 2016; Sun et al., 2016; Yoon et al., 2016; Barzegar et al., 2017; Ebrahimi and Rajaee, 2017; Wunsch et al., 2018; Zhang et al., 2018; Chen et al., 2019; Lee et al., 2019; Jeong et al., 2019 and the references therein). Most of these studies used methods like autoregressive integrated moving average (ARIMA), Artificial Neural Network (ANN), hybrid-ANN, Adaptive neuro-fuzzy inference system (ANFIS), genetic programming (GP), Support Vector Machine (SVM) and nonlinear auto-regressive exogenous model-based (NARX), long-short term memory (LSTM) model few others using a wide range of frequency and temporal data on past GWLs, satellite observations derived groundwater storage (GWS), Normalized difference vegetation index (NDVI)), meteorological variables, river discharge, variables of groundwater use, few dummy variables to simulate and/or predict GWLs. In a study by Yoon et al. (2011), ANN and SVM models were used using tide level, precipitation, and past GWLs as inputs to predict GWL fluctuations at a South Korean coastal aquifer. They reported that precipitation and tide levels are the most important input variables, and SVM performs better than ANN. Furthermore, the ability of GP, ANFIS, ANN, SVM, and ARIMA methods was evaluated by Shiri et al. (2013) in predicting GWL in Korea. The results suggest that the performance of GP is better than others. Using hybrid ANN with preprocessing approach Sahoo et al. (2017) predicted GWL change in some of the agriculture alluvial aquifers of the USA. Another recent study (Jeong et al., 2019) reported that NARX and LSTM methods provide good accuracy in predicting water level of two observation wells in the Korean peninsula using preprocessed climatic variables (temperature, precipitation, humidity, sunshine hours, and atmospheric pressure) that potentially affect GWL through changing the evapotranspiration and recharge. Zhang et al. (2018) identified stressed aquifer conditions by comparing the observed*

*and estimated GWL with LSTM using precipitation, temperature, water diversion, and evaporation as input. A recent study by Mukherjee and Ramachandran (2018) simulated GWLs for a small number (n = 5) of in-situ observation wells in India using Linear Regression Model (LRM), Artificial Neural Network (ANN), and Support Vector Regression (SVR) using Gravity Recovery and Climate Experiment (GRACE) derived terrestrial water storage (TWS) change and meteorological variables. However, the above-mentioned studies (including studies on India and Bangladesh) are mainly small-scale studies, and due to the small number of observation wells, they are unable to characterize the spatial variability in model performances extensively. Furthermore, the temporal extent of the studies on India and Bangladesh is often short (e.g., Mukherjee and Ramachandran (2018) considered the time period from 2005 to 2018). Hence the predictions are based on the short-term trends of dependent variables and do not consider the long-term variability. Moreover, using a combination of physically-based modeling and deep convolutional neural network (CNN), Sun et al. (2019) matched the GRACE based and simulated (by a land surface model as inputs) terrestrial water storage anomalies (TWSA). They further compared the calculated in-situ GWS (using specific yields and in-situ GWLs) with the variation between the observed and simulated model values and found a good correlation. However, this study does not use in-situ GWLs as model input and mainly based on the satellite observations and land surface model outputs. Moreover, a recent study by Yadav et al. (2020) used ANN and SVM on preprocessed data on GWL, precipitation, Southern Oscillation Index, Northern Oscillation Index, Niño3, and population as input to predict GWL in the urban areas of Bengaluru, India. They also discussed the significant impact of population growth in GWL estimation and prediction in urban areas in India (Yadav et al., 2020)."*

We further added on the originality of our study,

*"The previous studies, as well as the studies on Bangladesh and India, are mostly based on a small spatial and a short temporal extent. Furthermore, to our knowledge, none of the studies have considered the spatial and depth-wise performance variability of machine learning models in predicting GWL. The originality of this study lies in addressing some critical aspects which were not included in the previous studies. Firstly, to understand the spatial variability in machine learning-based model performances, we have considered a large network of monitoring wells (n = 2303) from 1985 to 2015 to simulate GWLs in the IGBM. Secondly, considering the variable*

*patterns of groundwater abstraction, we showed the significance of well depth (intake depth of the observation wells) information in GWL modeling using machine learning. Thirdly, we used meteorological variables exclusively to simulate in-situ GWL. Fourthly, based on dominance analysis and outputs from the machine learning models, we investigated the most influential basin specific predictor(s) (both natural and human-induced) in GWL modeling."*

**Reference**

Barzegar, R., Fijani, E., Asghari Moghaddam, A. and Tziritis, E.: Forecasting of groundwater level fluctuations using ensemble hybrid multi-wavelet neural network-based models, Sci. Total Environ., 599–600, 20–31, doi:10.1016/j.scitotenv.2017.04.189, 2017.

Bhanja, S. N., Malakar, P., Mukherjee, A., Rodell, M., Mitra, P. and Sarkar, S.: Using Satellite-Based Vegetation Cover as Indicator of Groundwater Storage in Natural Vegetation Areas, Geophys. Res. Lett., 46(14), 8082–8092, doi:10.1029/2019GL083015, 2019b.

Chen, H., Zhang, W., Nie, N. and Guo, Y.: Long-term groundwater storage variations estimated in the Songhua River Basin by using GRACE products, land surface models, and in-situ observations, Sci. Total Environ., 649, 372–387, doi:10.1016/j.scitotenv.2018.08.352, 2019.

Coulibaly, P., Anctil, F., Aravena, R. and Bobée, B.: Artificial neural network modeling of water table depth fluctuations, Water Resour. Res., 37(4), 885–896, doi:10.1029/2000WR900368, 2001.

Ebrahimi, H. and Rajaee, T.: Simulation of groundwater level variations using wavelet combined with neural network, linear regression and support vector machine, Glob. Planet. Change, 148, 181–191, doi:10.1016/j.gloplacha.2016.11.014, 2017.

Feng, S., Kang, S., Huo, Z., Chen, S. and Mao, X.: Neural networks to simulate regional ground water levels affected by human activities, Ground Water, 46(1), 80–90, doi:10.1111/j.1745-6584.2007.00366.x, 2008.

Jeong, J. and Park, E.: Comparative applications of data-driven models representing water table fluctuations, J. Hydrol., 572(March), 261–273, doi:10.1016/j.jhydrol.2019.02.051, 2019.

Lee, S., Lee, K. K. and Yoon, H.: Using artificial neural network models for groundwater level forecasting and assessment of the relative impacts of influencing factors, Hydrogeol. J., 27(2), 567–579, doi:10.1007/s10040-018-1866-3, 2019.

Malakar, P., Mukherjee, A., Sarkar, S.: Potential Application of Advanced Computational Techniques in Prediction of Groundwater Resource of India, in: Mukherjee, A. (Ed.), Groundwater of South Asia. Springer Singapore, Singapore, pp. 643–655, 2018.

Mukherjee, A. and Ramachandran, P.: Prediction of GWL with the help of GRACE TWS for unevenly spaced time series data in India : Analysis of comparative performances of SVR, ANN and LRM, J. Hydrol., 558(October 2008), 647–658, doi:10.1016/j.jhydrol.2018.02.005, 2018.

Nayak, P. C., Satyaji Rao, Y. R. and Sudheer, K. P.: Groundwater level forecasting in a shallow aquifer using artificial neural network approach, Water Resour. Manag., 20(1), 77–90, doi:10.1007/s11269-006-4007-z, 2006.

Nourani, V. and Mousavi, S.: Spatiotemporal groundwater level modeling using hybrid artificial intelligence-meshless method, J. Hydrol., 536, 10–25, doi:10.1016/j.jhydrol.2016.02.030, 2016.

Nury, A. H., Hasan, K. and Alam, M. J. Bin: Comparative study of wavelet-ARIMA and wavelet-ANN models for temperature time series data in northeastern Bangladesh, J. King Saud Univ. - Sci., 29(1), 47–61, doi:10.1016/j.jksus.2015.12.002, 2017.

Sahoo, S., Russo, T. A., Elliott, J. and Foster, I.: Machine learning algorithms for modeling groundwater level changes in agricultural regions of the U.S., Water Resour. Res., 53(5), 3878–3895, doi:10.1002/2016WR019933, 2017.

Shiri, J., Kisi, O., Yoon, H., Lee, K. K. and Hossein Nazemi, A.: Predicting groundwater level fluctuations with meteorological effect implications-A comparative study among soft computing techniques, Comput. Geosci., 56, 32–44, doi:10.1016/j.cageo.2013.01.007, 2013.

Sun, A. Y.: Predicting groundwater level changes using GRACE data, Water Resour. Res., 49(9), 5900–5912, doi:10.1002/wrcr.20421, 2013.

Sun, A. Y., Scanlon, B. R., Zhang, Z., Walling, D., Bhanja, S. N., Mukherjee, A. and Zhong, Z.: Combining Physically Based Modeling and Deep Learning for Fusing GRACE Satellite Data: Can We Learn From Mismatch?, Water Resour. Res., doi:10.1029/2018WR023333, 2019.

Sun, Y., Wendi, D., Kim, D. E. and Liong, S. Y.: Technical note: Application of artificial neural networks in groundwater table forecasting-a case study in a Singapore swamp forest, Hydrol. Earth Syst. Sci., 20(4), 1405–1412, doi:10.5194/hess-20-1405-2016, 2016.

Wunsch, A., Liesch, T. and Broda, S.: Forecasting groundwater levels using nonlinear autoregressive networks with exogenous input (NARX), J. Hydrol., 567, 743–758, doi:10.1016/j.jhydrol.2018.01.045, 2018.

Yadav, B., Gupta, P. K., Patidar, N. and Himanshu, S. K.: Ensemble modelling framework for groundwater level prediction in urban areas of India, Sci. Total Environ., 135539, doi:10.1016/j.scitotenv.2019.135539, 2019.

Yoon, H., Jun, S. C., Hyun, Y., Bae, G. O. and Lee, K. K.: A comparative study of artificial neural networks and support vector machines for predicting groundwater levels in a coastal aquifer, J. Hydrol., 396(1–2), 128–138, doi:10.1016/j.jhydrol.2010.11.002, 2011.

Yoon, H., Hyun, Y., Ha, K., Lee, K. K. and Kim, G. B.: A method to improve the stability and accuracy of ANN- and SVM-based time series models for long-term groundwater level predictions, Comput. Geosci., 90, 144–155, doi:10.1016/j.cageo.2016.03.002, 2016.

Zhang, J., Zhu, Y., Zhang, X., Ye, M. and Yang, J.: Developing a Long Short-Term Memory (LSTM) based model for predicting water table depth in agricultural areas, J. Hydrol., 561(April), 918–929, doi:10.1016/j.jhydrol.2018.04.065, 2018.

**Rev 1. Minor Comment 1:** Figure S14. There are errors for the label of y-axis."SVM A" should not be followed by "ANN B" and "ANN C"

Reply: We thank the reviewer for noticing the typo. Following the reviewer's concern, we have corrected the typo in Figure S14.

[Figure]

*"Figure S14. Distribution of observation well counts with r, NSE RMSE$_n$ for the Brahmaputra basin."*

**Rev 1. Minor Comment 2:** Figure S3. I suggest the flowchart can be moved to the main text.

Reply: We thank the reviewer for the suggestions. Following the reviewer's comment, we have moved the flowchart to the main text as Figure 2.

**Rev 1. Minor Comment 3:** The manuscript analyzed the influence of population and groundwater withdrawal. These two variables may be related, this need to be clarified, and what are the purpose for groundwater abstraction?

Reply: We would like to thank the reviewer for this concern. We agree with the reviewer that population and groundwater withdrawal are interlinked parameters in some aspects. For example, assuming per capita groundwater withdrawal for domestic purposes is nearly equal, a net rise in population is directly proportional to the rise in groundwater withdrawal for domestic purposes. However, domestic withdrawal is limited to only ~4-8% of the total groundwater withdrawal in the basin, while irrigation-linked groundwater withdrawal contributes more than 90%. Irrigation strategies are shifting from flood irrigation to drip and sprinkler based irrigation systems, and this would continue in the near future. Thus, the water withdrawal for irrigation purposes (being the highest consumer of groundwater) is not directly linked to the population increase of the study area. This is the reason we have considered two separate parameters for designing this study.

"*We considered both the population and groundwater withdrawal as input parameters in the dominance analysis. Although these two parameters seem to be interlinked, however, in reality, they are not directly related in the IGBM basin. For example, assuming per capita groundwater withdrawal for domestic purposes is not changing over the years, a net rise in population is directly proportional to the rise in groundwater withdrawal for domestic purposes. However, domestic withdrawal is limited to only ~4-8% of the total groundwater withdrawal in the basin (Sharma et al., 2008; CGWB, 2019). Irrigation-linked groundwater withdrawal contributes more than 90% throughout the basin (Sharma et al., 2008; CGWB, 2019). Irrigation strategies are shifting from flood irrigation to drip and sprinkler based irrigation systems, and this would continue in the near future. Thus, the water withdrawal for irrigation purposes (being the highest consumer of groundwater) is not directly linked to population increase; rather, it is dependent upon the irrigation strategies used (Bhanja et al., 2017a).*"

**References**

Sharma, B. R., Amarasinghe, U. A., & Sikka, A.: Indo-Gangetic river basins: Summary situation analysis. New Delhi, India: International Water Management Institute. Pp14, 2008

Central Ground Water Board: Dynamic groundwater resources, India, 2017, Govt. of India, Ministry of Water Resources, Faridabad, 306 pp., 2019

Bhanja, S. N., Mukherjee, A., Rodell, M., Wada, Y., Chattopadhyay, S., Velicogna, I., Pangaluru, K. and Famiglietti, J. S.: Groundwater rejuvenation in parts of India influenced by water-policy change implementation, Sci. Rep., 7(1), 7453, doi:10.1038/s41598-017-07058-2, 2017a.

---

## Author Comment (AC2) · 22 Oct 2020

*"Importance of spatial and depth-dependent drivers in groundwater level modeling through machine learning"* by Pragnaditya Malakar, Abhijit Mukherjee, Soumendra N. Bhanja, Dipankar Saha, Ranjan Kumar Ray, Sudeshna Sarkar, Anwar Zahid

**Reviewer #2:** Groundwater is an important source of water, in particular for the transboundary areas of IGBM Rivers. This study used a liner regression approach based on dominance analysis and machine learning methods to identify the spatial and depth-wise drivers based on a large network of ground-based observations. Some interesting conclusions are found by the authors, including e.g. the groundwater level change is primarily influenced by abstraction and population in most of the IGBM; the machine learning methods can well simulate the groundwater level and the performance decreases from shallow to deep observation wells. The conclusions can be useful for groundwater management in the IGBM areas. However, the quality of this manuscript is not good enough for publication in HESS. The detailed comments are shown as the following.

Reply: We thank the reviewer for his/her review and support for the general intent of the paper. We appreciate that the appended comments are helpful and intended to improve the manuscript. We have addressed the reviewer's comments and done a complete major revision of the manuscript. Doing so, we have added details on the method, results, and discussion, rewritten several sections, and added a few new figures, which we believe have greatly improved the manuscript.

**Highlights of the revision:**

In this revision, we have

a)  Included a detailed discussion of the prevailing methods, differences between our study and previous studies.
b)  Highlighted the originality of the present study
c)  Shown the representative groundwater level time series for different basins
d)  Discussed the interdependence of the variables and its relation to dominance analysis
e)  Included the rationale for using ANN and SVM in the study
f)  Discussed possible limitation of using machine learning methods in groundwater level modeling

g) Explained the possible reasons for relatively large deviations in the Indus basin.

**Rev 2. Comment 1:** Machine learning methods are popular over the years. The authors gave an introduction to machine learning methods used in GWL. I expect that more prevailing methods should be mentioned in the introduction instead of ANN and SVR. I also expect a comparison of these prevailing methods.

Reply: We thank the reviewer for the comment. In recent times, machine learning-based methods have been widely used to simulate and predict groundwater levels across the world. Most of these studies used methods like autoregressive integrated moving average (ARIMA), Artificial Neural Network (ANN), hybrid-ANN, Adaptive neuro-fuzzy inference system (ANFIS), genetic programming (GP), Support Vector Machine (SVM) and nonlinear auto-regressive exogenous model-based (NARX), long-short term memory (LSTM) model few others using a wide range of frequency and temporal data on past GWLs, satellite observations derived groundwater storage (GWS), Normalized difference vegetation index (NDVI)), meteorological variables, river discharge, variables of groundwater use, few dummy variables to simulate and/or predict GWLs. However, most studies (including studies on India and Bangladesh) are mainly small-scale studies, and due to the small number of observation wells, they are unable to characterize the spatial variability in model performances extensively. Moreover, the temporal extent of the studies on India and Bangladesh is often short. Hence the predictions are based on the short-term trends of dependent variables and do not consider the long-term variability. Furthermore, to our knowledge, none of the studies have considered the spatial and depth-wise performance variability of machine learning models in predicting GWL. The originality of this study lies in addressing some critical aspects which were not included in the previous studies. Firstly, to understand the spatial variability in machine learning-based model performances, we have considered a large network of monitoring wells (n = 2303) from 1985 to 2015 to simulate GWLs in the IGBM. Secondly, considering the variable depth-wise patterns of groundwater abstraction, we showed the significance of well depth (intake depth of the observation wells) information in GWL modeling using machine learning. Thirdly, we used meteorological variables exclusively to simulate in-situ GWLs. Fourthly, based on dominance analysis and outputs from the machine learning models, we investigated the most influential basin specific predictor(s) (both natural and human-induced) in GWL modeling. Following the reviewer's suggestion, we have modified the text, including the

findings of the previous studies, and added new text to show the differences and originality of the present study.

Following the suggestion of the reviewer, we have added other prevailing methods in the introduction section. We further modified the text regarding the comparison of the prevailing methods and added new text to show the difference and originality of the study.

*"Over the years, the simplistic approach and acceptable results of the machine learning (ML) methods are preferred when the underlying physical system is not well understood. Sometimes, decoding the physical system becomes much more complex due to interactions and feedbacks between multiple processes. GWL modeling based on ML has the unique ability to find the likely relationships between GWL and controlling hydro-climatic-anthropogenic variables without constructing knowledge-driven conceptual or physically-based models. Therefore, researchers have studied the performance of ML methods for GWL modeling in India and Bangladesh (Nayak et al., 2006; Nury et al., 2017; Malakar et al., 2018; Mukherjee and Ramachandran, 2018; Bhanja et al., 2019b; Sun et al., 2019; Yadav et al., 2020; and the references therein) and other parts of the world (Coulibaly et al., 2001; Feng et al., 2008; Sun, 2013; Nourani and Mousavi, 2016; Sun et al., 2016; Yoon et al., 2016; Barzegar et al., 2017; Ebrahimi and Rajaee, 2017; Wunsch et al., 2018; Zhang et al., 2018; Chen et al., 2019; Lee et al., 2019; Jeong et al., 2019 and the references therein). Most of these studies used methods like autoregressive integrated moving average (ARIMA), Artificial Neural Network (ANN), hybrid-ANN, Adaptive neuro-fuzzy inference system (ANFIS), genetic programming (GP), Support Vector Machine (SVM) and nonlinear auto-regressive exogenous model-based (NARX), long-short term memory (LSTM) model few others using a wide range of frequency and temporal data on past GWLs, satellite observations derived groundwater storage (GWS), Normalized difference vegetation index (NDVI)), meteorological variables, river discharge, variables of groundwater use, few dummy variables to simulate and/or predict GWLs. In a study by Yoon et al. (2011), ANN and SVM models were used using tide level, precipitation, and past GWLs as inputs to predict GWL fluctuations at a South Korean coastal aquifer. They reported that precipitation and tide levels are the most important input variables, and SVM performs better than ANN. Furthermore, the ability of GP, ANFIS, ANN, SVM, and ARIMA methods was evaluated by Shiri et al. (2013) in predicting GWL in Korea. The results suggest that the performance of GP is better than others. Using hybrid ANN with preprocessing*

*approach Sahoo et al. (2017) predicted GWL change in some of the agriculture alluvial aquifers of the USA. Another recent study (Jeong et al., 2019) reported that NARX and LSTM methods provide good accuracy in predicting the water level of two observation wells in the Korean peninsula using preprocessed climatic variables (temperature, precipitation, humidity, sunshine hours, and atmospheric pressure) that potentially affect GWL through changing the evapotranspiration and recharge. Zhang et al. (2018) identified stressed aquifer conditions by comparing the observed and estimated GWL with LSTM using precipitation, temperature, water diversion, and evaporation as input. A recent study by Mukherjee and Ramachandran (2018) simulated GWLs for a small number (n = 5) of in-situ observation wells in India using Linear Regression Model (LRM), Artificial Neural Network (ANN), and Support Vector Regression (SVR) using Gravity Recovery and Climate Experiment (GRACE) derived terrestrial water storage (TWS) change and meteorological variables. However, the above-mentioned studies (including studies on India and Bangladesh) are mainly small-scale studies, and due to the small number of observation wells, they are unable to characterize the spatial variability in model performances extensively. Furthermore, the temporal extent of the studies on India and Bangladesh is often short (e.g., Mukherjee and Ramachandran (2018) considered the time period from 2005 to 2018). Hence the predictions are based on the short-term trends of dependent variables and do not consider the long-term variability. Moreover, using a combination of physically-based modeling and deep convolutional neural network (CNN), Sun et al. (2019) matched the GRACE based and simulated (by a land surface model as inputs) terrestrial water storage anomalies (TWSA). They further compared the calculated in-situ GWS (using specific yields and in-situ GWLs) with the variation between the observed and simulated model values and found a good correlation. However, this study does not use in-situ GWLs as model input and mainly based on the satellite observations and land surface model outputs. Moreover, a recent study by Yadav et al. (2020) used ANN and SVM on preprocessed data on GWL, precipitation, Southern Oscillation Index, Northern Oscillation Index, Niño3, and population as input to predict GWL in the urban areas of Bengaluru, India. They also discussed the significant impact of population growth in GWL estimation and prediction in urban areas in India (Yadav et al., 2020)."*

We further added on the originality of our study,

*"The previous studies, as well as the studies on Bangladesh and India, are mostly based on a small spatial and a short temporal extent. Furthermore, to our knowledge, none of the studies have considered the spatial and depth-wise performance variability of machine learning models in predicting GWL. The originality of this study lies in addressing some critical aspects which were not included in the previous studies. Firstly, to understand the spatial variability in machine learning-based model performances, we have considered a large network of monitoring wells (n = 2303) from 1985 to 2015 to simulate GWLs in the IGBM. Secondly, considering the variable patterns of groundwater abstraction, we showed the significance of well depth (intake depth of the observation wells) information in GWL modeling using machine learning. Thirdly, we used meteorological variables exclusively to simulate in-situ GWL. Fourthly, based on dominance analysis and outputs from the machine learning models, we investigated the most influential basin specific predictor(s) (both natural and human-induced) in GWL modeling."*

**Reference**

Barzegar, R., Fijani, E., Asghari Moghaddam, A. and Tziritis, E.: Forecasting of groundwater level fluctuations using ensemble hybrid multi-wavelet neural network-based models, Sci. Total Environ., 599–600, 20–31, doi:10.1016/j.scitotenv.2017.04.189, 2017.

Bhanja, S. N., Malakar, P., Mukherjee, A., Rodell, M., Mitra, P. and Sarkar, S.: Using Satellite-Based Vegetation Cover as Indicator of Groundwater Storage in Natural Vegetation Areas, Geophys. Res. Lett., 46(14), 8082–8092, doi:10.1029/2019GL083015, 2019b.

Chen, H., Zhang, W., Nie, N. and Guo, Y.: Long-term groundwater storage variations estimated in the Songhua River Basin by using GRACE products, land surface models, and in-situ observations, Sci. Total Environ., 649, 372–387, doi:10.1016/j.scitotenv.2018.08.352, 2019.

Coulibaly, P., Anctil, F., Aravena, R. and Bobée, B.: Artificial neural network modeling of water table depth fluctuations, Water Resour. Res., 37(4), 885–896, doi:10.1029/2000WR900368, 2001.

Ebrahimi, H. and Rajaee, T.: Simulation of groundwater level variations using wavelet combined with neural network, linear regression and support vector machine, Glob. Planet. Change, 148, 181–191, doi:10.1016/j.gloplacha.2016.11.014, 2017.

Feng, S., Kang, S., Huo, Z., Chen, S. and Mao, X.: Neural networks to simulate regional ground water levels affected by human activities, Ground Water, 46(1), 80–90, doi:10.1111/j.1745-6584.2007.00366.x, 2008.

Jeong, J. and Park, E.: Comparative applications of data-driven models representing water table fluctuations, J. Hydrol., 572(March), 261–273, doi:10.1016/j.jhydrol.2019.02.051, 2019.

Lee, S., Lee, K. K. and Yoon, H.: Using artificial neural network models for groundwater level forecasting and assessment of the relative impacts of influencing factors, Hydrogeol. J., 27(2), 567–579, doi:10.1007/s10040-018-1866-3, 2019.

Malakar, P., Mukherjee, A., Sarkar, S.: Potential Application of Advanced Computational Techniques in Prediction of Groundwater Resource of India, in: Mukherjee, A. (Ed.), Groundwater of South Asia. Springer Singapore, Singapore, pp. 643–655, 2018.

Mukherjee, A. and Ramachandran, P.: Prediction of GWL with the help of GRACE TWS for unevenly spaced time series data in India : Analysis of comparative performances of SVR, ANN and LRM, J. Hydrol., 558(October 2008), 647–658, doi:10.1016/j.jhydrol.2018.02.005, 2018.

Nayak, P. C., Satyaji Rao, Y. R. and Sudheer, K. P.: Groundwater level forecasting in a shallow aquifer using artificial neural network approach, Water Resour. Manag., 20(1), 77–90, doi:10.1007/s11269-006-4007-z, 2006.

Nourani, V. and Mousavi, S.: Spatiotemporal groundwater level modeling using hybrid artificial intelligence-meshless method, J. Hydrol., 536, 10–25, doi:10.1016/j.jhydrol.2016.02.030, 2016.

Nury, A. H., Hasan, K. and Alam, M. J. Bin: Comparative study of wavelet-ARIMA and wavelet-ANN models for temperature time series data in northeastern Bangladesh, J. King Saud Univ. - Sci., 29(1), 47–61, doi:10.1016/j.jksus.2015.12.002, 2017.

Sahoo, S., Russo, T. A., Elliott, J. and Foster, I.: Machine learning algorithms for modeling groundwater level changes in agricultural regions of the U.S., Water Resour. Res., 53(5), 3878–3895, doi:10.1002/2016WR019933, 2017.

Shiri, J., Kisi, O., Yoon, H., Lee, K. K. and Hossein Nazemi, A.: Predicting groundwater level fluctuations with meteorological effect implications-A comparative study among soft computing techniques, Comput. Geosci., 56, 32–44, doi:10.1016/j.cageo.2013.01.007, 2013.

Sun, A. Y.: Predicting groundwater level changes using GRACE data, Water Resour. Res., 49(9), 5900–5912, doi:10.1002/wrcr.20421, 2013.

Sun, A. Y., Scanlon, B. R., Zhang, Z., Walling, D., Bhanja, S. N., Mukherjee, A. and Zhong, Z.: Combining Physically Based Modeling and Deep Learning for Fusing GRACE Satellite Data: Can We Learn From Mismatch?, Water Resour. Res., doi:10.1029/2018WR023333, 2019.

Sun, Y., Wendi, D., Kim, D. E. and Liong, S. Y.: Technical note: Application of artificial neural networks in groundwater table forecasting-a case study in a Singapore swamp forest, Hydrol. Earth Syst. Sci., 20(4), 1405–1412, doi:10.5194/hess-20-1405-2016, 2016.

Wunsch, A., Liesch, T. and Broda, S.: Forecasting groundwater levels using nonlinear autoregressive networks with exogenous input (NARX), J. Hydrol., 567, 743–758, doi:10.1016/j.jhydrol.2018.01.045, 2018.

Yadav, B., Gupta, P. K., Patidar, N. and Himanshu, S. K.: Ensemble modelling framework for groundwater level prediction in urban areas of India, Sci. Total Environ., 135539, doi:10.1016/j.scitotenv.2019.135539, 2019.

Yoon, H., Jun, S. C., Hyun, Y., Bae, G. O. and Lee, K. K.: A comparative study of artificial neural networks and support vector machines for predicting groundwater levels in a coastal aquifer, J. Hydrol., 396(1–2), 128–138, doi:10.1016/j.jhydrol.2010.11.002, 2011.

Yoon, H., Hyun, Y., Ha, K., Lee, K. K. and Kim, G. B.: A method to improve the stability and accuracy of ANN- and SVM-based time series models for long-term groundwater level predictions, Comput. Geosci., 90, 144–155, doi:10.1016/j.cageo.2016.03.002, 2016.

Zhang, J., Zhu, Y., Zhang, X., Ye, M. and Yang, J.: Developing a Long Short-Term Memory (LSTM) based model for predicting water table depth in agricultural areas, J. Hydrol., 561(April), 918–929, doi:10.1016/j.jhydrol.2018.04.065, 2018.

**Rev 2. Comment 2:** From the manuscript, it is difficult to see the originality of this study. For me, the only originality might be the use of a large network of monitoring wells to identify the spatial and depth-wise drivers.

Reply: We thank the reviewer for the comment. Over the years, a host of machine learning methods is applied in groundwater level prediction and simulation worldwide. However, to our knowledge, the previous studies have not considered the spatial and depth-wise performance variability of machine learning models in simulating GWL. The originality of this article lies in addressing these critical aspects. Following the reviewer's comment, we have highlighted the originality of our study. We added,

*"The previous studies, as well as the studies on Bangladesh and India, are mostly based on a small spatial and a short temporal extent. Furthermore, to our knowledge, none of the studies have considered the spatial and depth-wise performance variability of machine learning models in predicting GWL. The originality of this study lies in addressing some critical aspects that were not included in the previous studies. Firstly, to understand the spatial variability in machine learning-based model performances, we have considered a large network of monitoring wells (n = 2303) from 1985 to 2015 to simulate GWLs in the IGBM. Secondly, considering the variable patterns of groundwater abstraction, we showed the significance of well depth (intake depth of the observation wells) information in GWL modeling using machine learning. Thirdly, we used meteorological variables exclusively to simulate in-situ GWL. Fourthly, based on dominance analysis and outputs from the machine learning models, we investigated the most influential basin specific predictor(s) (both natural and human-induced) in GWL modeling."*

**Rev 2. Comment 3:** Line 120: Although a large network of monitoring wells was used, the time resolution is rather coarse. Also can the authors show us the time series of monitored water levels?

Reply: We thank the reviewer for the comment. We would like to mention that this is the best possible dataset in terms of spatial resolution available in the region to date. We agree with the reviewer that the temporal resolution of the data is a little coarse; however, we could not find any other data with better temporal resolution covering the entire study area.

Based on the reviewer's suggestion, we have inserted the time series plots of groundwater levels from multiple locations. Since the time series of all the 2303 monitoring wells was not possible to show, following the reviewer's comment, we have shown the 100 randomly selected observation wells in each of the basins with most records in the supplementary information.

[Figure]

*Figure S19. Borehole hydrographs (GWL depths below surface) of 100 representative monitoring wells with most records in the Indus basin.*

[Figure]

*Figure S20. Borehole hydrographs (GWL depths below surface) of 100 representative monitoring wells with most records in the Ganges basin.*

[Figure]

*Figure S21. Borehole hydrographs (GWL depths below surface) of 100 representative monitoring wells with most records in the Brahmaputra basin.*

[Figure]

*Figure S22. Borehole hydrographs (GWL depths below surface) of 100 representative monitoring wells with most records in the Meghna basin.*

**Rev 2. Comment 4:** For the dominance analysis, the independent variables seem dependent, such as groundwater withdrawals and population, temperature and potential evapotranspiration. Will this affect the results of dominance analysis?

Reply: We would like to thank the reviewer for this concern. We agree with the reviewer that population and groundwater withdrawal are interlinked parameters in some aspects. For example, assuming per capita groundwater withdrawal for domestic purposes is nearly equal, a net rise in population is directly proportional to the rise in groundwater withdrawal for domestic purposes. However, domestic withdrawal is limited to only ~4-8% of the total groundwater withdrawal in the basin, while irrigation-linked groundwater withdrawal contributes more than 90%. Irrigation strategies are shifting from flood irrigation to drip and sprinkler based irrigation systems, and this would continue in the near future. Thus, the water withdrawal for irrigation purposes (being the highest consumer of groundwater) is not directly linked to the population increase of the study area. This is the reason we have considered two separate parameters for designing this study.

"*We considered both the population and groundwater withdrawal as input parameters in the dominance analysis. Although these two parameters seem to be interlinked, however, in reality, they are not directly related in the IGBM basin. For example, assuming per capita groundwater withdrawal for domestic purposes is not changing over the years, a net rise in population is directly proportional to the rise in groundwater withdrawal for domestic purposes. However, domestic withdrawal is limited to only ~4-8% of the total groundwater withdrawal in the basin (Sharma et al., 2008; CGWB, 2019). Irrigation-linked groundwater withdrawal contributes more than 90% throughout the basin (Sharma et al., 2008; CGWB, 2019). Irrigation strategies are shifting from flood irrigation to drip and sprinkler based irrigation systems, and this would continue in the near future. Thus, the water withdrawal for irrigation purposes (being the highest consumer of groundwater) is not directly linked to population increase; rather, it is dependent upon the irrigation strategies used (Bhanja et al., 2017a).*"

The temperature could be considered as a proxy of water uses for irrigation. We also agree with the reviewer that potential evapotranspiration is controlled by ambient temperature. Furthermore, in addition to temperature, PET (used in the study) is calculated from variables such as net radiation at crop surface, wind speed, soil heat flux, and vapour pressure using the Penman-Monteith formula. Thus, these variables influencing the potential evapotranspiration are significant components of the regional hydrological cycle. Hence, we have included PET in our analyses.

We included in the text,

*"Temperature could be considered as a proxy of water uses for irrigation (Sun, 2013). Furthermore, potential evapotranspiration (PET) dependents on temperature, net radiation at crop surface, wind speed, soil heat flux, and vapour pressure (Ekström et al., 2007; Harris et al., 2020). PET has been included in the analysis since these variables are significant components of the regional hydrological cycle."*

Furthermore, the dominance analysis (DA) computes the coefficient of determination ($R^2$) in multiple regression of all possible predictor combinations. The relative importance of the predictor variables is determined by the DA through a pair-wise comparisons of all predictors in the multiple regression model as they relate to an outcome variable. The DA has been developed specifically

to address the issue of multicollinearity between predictors and provide the relative importance not affected by the multicollinearity between the predictors.

Following the reviewer's comment, we have added a brief description.

**2.4 Dominance analysis**

*"Yearly precipitation, temperature, groundwater withdrawals, population, and potential evapotranspiration for the IGBM and each of sub-basins were taken as the predictor variable to understand their relationship with the outcome variable GWLs. However, the predictor variables used in the study could be interrelated to some degree. Thus it is hard to assess the importance of each predictor if there exists a high degree of multicollinearity within the predictors. The DA has been developed specifically to address the issues and provide the relative not affected by the multicollinearity between the predictors. (Tighe et al., 2014).*

*The dominance analysis (DA) computes the coefficient of determination ($R^2$) in multiple regression of all possible predictor combinations (Azen and Budescu, 2006; Budescu, 1993; Thomas and Famiglietti, 2019). The relative importance of the predictor variables is determined by the DA through a pair-wise comparisons of all predictors in the multiple regression model as they relate to an outcome variable (Budescu, 1993; Tighe et al., 2014). The DA method partition coefficient of determination ($R^2$) of the overall multiple regression into "shares". These shares are attributable to each of the predictors (Braun et al., 2019) and identify the predictors that are more or less important or dominant than others. Here, the conditional dominance of the variables for p-1 sub-models is performed (where p is the numeric value of total sub-models) (Thomas and Famiglietti, 2019). A comprehensive narrative on the dominance analysis can be found in Budescu (1993) and Azen and Budescu (2006)."*

**References**

Azen, R. and Budescu, D. V.: Comparing predictors in multivariate regression models: An extension of dominance analysis, J. Educ. Behav. Stat., 31(2), 157–180, doi:10.3102/10769986031002157, 2006.

Bhanja, S. N., Mukherjee, A., Rodell, M., Wada, Y., Chattopadhyay, S., Velicogna, I., Pangaluru, K. and Famiglietti, J. S.: Groundwater rejuvenation in parts of India influenced by water-policy change implementation, Sci. Rep., 7(1), 7453, doi:10.1038/s41598-017-07058-2, 2017a.

Budescu, D. V.: Dominance analysis: A new approach to the problem of relative importance of predictors in multiple regression, Psychol. Bull., 114(3), 542–551, doi:10.1037/0033-2909.114.3.542, 1993.

Central Ground Water Board: Dynamic groundwater resources, India, 2017, Govt. of India, Ministry of Water Resources, Faridabad, 306 pp., 2019

Ekström, M., Jones, P. D., Fowler, H. J., Lenderink, G., Buishand, T. A. and Conway, D.: Regional climate model data used within the SWURVE project projected changes in seasonal patterns and estimation of PET, Hydrol. Earth Syst. Sci., 11(3), 1069–1083, doi:10.5194/hess-11-1069-2007, 2007.

Harris, I., Osborn, T. J., Jones, P. and Lister, D.: Version 4 of the CRU TS monthly high-resolution gridded multivariate climate dataset, Sci. Data, 7(1), 1–18, doi:10.1038/s41597-020-0453-3, 2020.

Sharma, B. R., Amarasinghe, U. A., & Sikka, A.: Indo-Gangetic river basins: Summary situation analysis. New Delhi, India: International Water Management Institute. Pp14, 2008

Sun, A. Y.: Predicting groundwater level changes using GRACE data, Water Resour. Res., 49(9), 5900–5912, doi:10.1002/wrcr.20421, 2013.

Thomas, B. F. and Famiglietti, J. S.: Identifying Climate-Induced Groundwater Depletion in GRACE Observations, Sci. Rep., 9(1), doi:10.1038/s41598-019-40155-y, 2019.

Tighe, E. L. and Schatschneider, C.: A dominance analysis approach to determining predictor importance in third, seventh, and tenth grade reading comprehension skills, Read. Writ., 27(1), 101–127, doi:10.1007/s11145-013-9435-6, 2014.

**Rev 2. Comment 5:** Section 2.5: I am curious why the authors used two somewhat old-fashioned models including ANN and SVM. It is very easy to over-train these two types of models. I suggest the authors to use other models including LSTM.

Reply: We would like to thank the reviewer for the comment. We agree that ANN and SVM are used for quite a long time. These methods are seemed to be robust, widely accepted, and extensively used in the earth science discipline. We have mentioned the rationale for using ANN and SVM in our study. We added,

*"Despite many computationally intensive and suitable machine learning methods available in the literature, ANN and SVM are selected here since these are the two most popular and widely used methods in predicting GWL, and proven to provide very good results in predicting GWL worldwide and in the similar study areas. The previous studies, as well as the studies on Bangladesh and India, are mostly based on a small spatial and a short temporal extent. Furthermore, to our knowledge, none of the studies have considered the spatial and depth-wise performance variability of machine learning models in predicting GWL."*

**Rev 2. Comment 6:** Line 251: replace 'has' with 'have'

Reply: We thank the reviewer for noticing the typo. We have corrected the typo in the text.

We corrected,

*"Moreover, in general, the ML methods may have some weaknesses in modeling complex relationships regarding the low generalizability of the methods, risk of overtraining (Rajaee et al., 2019; Boutaghane et al., 2020)."*

**References**

Boutaghane, T. B. M. G. H.: Impact of training data size on the LSTM performances for rainfall – runoff modeling, Model. Earth Syst. Environ., 6(4), 2153–2164, doi:10.1007/s40808-020-00830-w, 2020.

Rajaee, T., Ebrahimi, H. and Nourani, V.: A review of the artificial intelligence methods in groundwater level modeling, J. Hydrol., 572(May 2018), 336–351, doi:10.1016/j.jhydrol.2018.12.037, 2019.

**Rev 2. Comment 7:** If the ML methods used in the study have some weakness regarding the low generalizability of the methods, risk of overtraining, why did the authors choose other machine learning methods?

Reply: We thank the reviewer for the comment. The statement of ML methods having low generalizability, risk of overtraining may be true to most of the other ML methods when modeling complex relationships. We mentioned this in Section 2.6 (Limitations, assumptions, and uncertainty) to highlight the possible drawback of using ML methods in general. Other methods, such as LSTM, are also subjected to these issues. In one of the recent studies, Boutaghane et al. (2020) reported that a poor training procedure of the LSTM for these situations leading to a bad generalization ability. In fact, they further added the possibility to be trapped in local minima with bad generalization in such complex relationships as rainfall-runoff is possible, even when using the advanced Adam algorithm with measures taken to avoid the overtraining problem (holdout method and dropout layer). We have modified the sentence as

*"Moreover, in general, most of the ML methods may have some weaknesses in modeling complex relationships regarding the low generalizability of the methods, risk of overtraining (Rajaee et al., 2019; Boutaghane et al., 2020)."*

Furthermore, we performed a sensitivity analysis using different model configurations and selected the model that provides better performances.

**References**

Boutaghane, T. B. M. G. H.: Impact of training data size on the LSTM performances for rainfall – runoff modeling, Model. Earth Syst. Environ., 6(4), 2153–2164, doi:10.1007/s40808-020-00830-w, 2020.

Rajaee, T., Ebrahimi, H. and Nourani, V.: A review of the artificial intelligence methods in groundwater level modeling, J. Hydrol., 572(May 2018), 336–351, doi:10.1016/j.jhydrol.2018.12.037, 2019.

**Rev 2. Comment 8:** Line 268: it seems to me that only half of the observation wells having correlations greater than 0.6 is not much.

Reply: We thank the reviewer for the comment. In this study, we performed a comparative analysis involving different models and different combinations of input parameters. Our findings suggest that using the best model, which is Model B (SVM, using GWL and meteorological variables as input) in the study, almost 76% of the observation wells show correlations greater than 0.6. Thus, in the concluding remark, we recommended using SVM with GWL and meteorological variables as input. We added,

*"Furthermore, model efficiency improves when climatic variables are included as input variables in addition to past GWLs into the system. Thus, the best performance in predicting GWL is found when SVM is used with GWL and meteorological variables as input. It is recommended to use SVM with GWL and meteorological variables as input in performing similar analysis."*

**Rev 2. Comment 9:** Line 328: it is expected that the ANN and SVM models have limitations in areas with higher groundwater abstraction.

Reply: We agree with the reviewer.

**Rev 2. Comment 10:** Figure 4: why large deviations in Indus?

Reply: We thank the reviewer for the comment. Figure 4 illustrates the comparative time series of the observed and simulated median groundwater levels for all the basin, sub-basin, and depth categories using ANN and SVM. The Indus basin is the most exploited basin in the study area. The shallow and deep observation wells are significantly influenced by a host of natural and anthropogenic drivers, resulting in a very complex hydrological condition. Hence, it is difficult to model the complex hydrological relationship. We have explained the probable reasons for large deviations in the Indus basin:

*"Indus basin is the most exploited basin in the study area (Figure 1c, Table S4). Thus, observation wells in the Indus basins are severely influenced by the local scale groundwater abstraction. Furthermore, irrigation return flow (Bhanja et al., 2019b) and inflow from canal leakage (Macdonald et al., 2015; Macdonald et al., 2016) may influence the shallow observation wells in the Indus basin. Moreover, the bulk of groundwater abstraction in the Indus basin is accounted for from the deep aquifers (which are often confined) through the deep irrigation wells in the agriculture regions (Girotto et al., 2017). Thus, the deep observation wells significantly influenced*

*by deep irrigational activity. Moreover, the deep confined aquifers are affected by the pumping with a much larger head decline at a larger area and the equilibrium time depending upon the nature of the confining bed (Alley et al., 1999; Castellazzi et al., 2016). Thus, the GWL changes in the Indus basin involves several natural and human-influenced drivers, resulting in a rather complex hydrological condition. Hence, it is difficult to model the complex hydrological relationship. More information on sub-surface geometry and local groundwater pumping is needed for a better understanding. Furthermore, due to the changes in the irrigational pattern (Figure S17, shallow vs. deep irrigational wells), we observed differential GWL hydrographs (Figure S19, 100 representative monitoring wells in Indus basin) in the training and testing stages of the machine learning modeling. This could also be a potential reason for relatively weak model performances in the Indus basin."*

**References**

Alley, W.M., Reilly, T.E., Franke, O.L.: Sustainability of groundwater resources. U.S. Geological Survey Circular 1186. http://pubs.usgs.gov/circ/circ1186/pdf/circ1186 .pdf (accessed September 10, 2020), 1999.

Bhanja, S. N., Mukherjee, A., Rangarajan, R., Scanlon, B. R., Malakar, P. and Verma, S.: Long-term groundwater recharge rates across India by in situ measurements, Hydrol. Earth Syst. Sci., 23(2), 711–722, doi:10.5194/hess-23-711-2019, 2019b.

Castellazzi, P., Martel, R., Galloway, D. L., Longuevergne, L. and Rivera, A.: Assessing Groundwater Depletion and Dynamics Using GRACE and InSAR: Potential and Limitations, Groundwater, 54(6), 768–780, doi:10.1111/gwat.12453, 2016.

Girotto, M., De Lannoy, G. J. M., Reichle, R. H., Rodell, M., Draper, C., Bhanja, S. N. and Mukherjee, A.: Benefits and pitfalls of GRACE data assimilation: A case study of terrestrial water storage depletion in India, Geophys. Res. Lett., 44(9), 4107–4115, doi:10.1002/2017GL072994, 2017.

MacDonald, A.M., Bonsor, H. C., Taylor, R., Shamsudduha, M., Burgess, W. G., Ahmed, K. M., Mukherjee, A., Zahid, A., Lapworth, D., Gopal, K., Rao, M. S., Moench, M., Bricker, S. H., Yadav, S. K., Satyal, Y., Smith, L., Dixit, A., Bell, R., van Steenbergen, F., Basharat, M., Gohar, M. S.,

Tucker, J., Calow, R. . C. and Maurice, L.: Groundwater resources in the Indo-Gangetic Basin: resilience to climate change and abstraction, 2015.

MacDonald, A. M., Bonsor, H. C., Ahmed, K. M., Burgess, W. G., Basharat, M., Calow, R. C., Dixit, A., Foster, S. S. D., Gopal, K., Lapworth, D. J., Lark, R. M., Moench, M., Mukherjee, A., Rao, M. S., Shamsudduha, M., Smith, L., Taylor, R. G., Tucker, J., Van Steenbergen, F. and Yadav, S. K.: Groundwater quality and depletion in the Indo-Gangetic Basin mapped from in situ observations, Nat. Geosci., 9(10), 762–766, doi:10.1038/ngeo2791, 2016.

**Rev 2. Comment 11:** Figure 6: how were the relative contributions calculated? Based on coefficient of determination?

Reply: We thank the reviewer for the comment. The dominance analysis (DA) computes the coefficient of determination ($R^2$) in multiple regression of all possible predictor combinations. The relative importance of the predictor variables is determined by the DA through a pair-wise comparisons of all predictors in the multiple regression model as they relate to an outcome variable. Following the reviewer's comments, we have added a brief description. We modified and added,

**2.4 Dominance analysis**

*"Yearly precipitation, temperature, groundwater withdrawals, population, and potential evapotranspiration for the IGBM and each of sub-basins were taken as the predictor variable to understand their relationship with the outcome variable GWLs. However, the predictor variables used in the study could be related to some degree. Thus it is hard to assess the importance of each predictor if there exists a high degree of multicollinearity within the predictors. The DA has been developed specifically to address the issues and provide the relative not affected by the multicollinearity between the predictors. (Tighe et al., 2014).*

*The dominance analysis (DA) computes the coefficient of determination ($R^2$) in multiple regression of all possible predictor combinations (Azen and Budescu, 2006; Budescu, 1993; Thomas and Famiglietti, 2019). The relative importance of the predictor variables is determined by the DA through a pair-wise comparisons of all predictors in the multiple regression model as they relate to an outcome variable (Budescu, 1993; Tighe et al., 2014). The DA method partition coefficient*

*of determination ($R^2$) of the overall multiple regression into "shares". These shares are attributable to each of the predictors (Braun et al., 2019) and identify the predictors that are more or less important or dominant than others. Here, the conditional dominance of the variables for p-1 sub-models is performed (where p is the numeric value of total sub-models) (Thomas and Famiglietti, 2019). A comprehensive narrative on the dominance analysis can be found in Budescu (1993) and Azen and Budescu (2006)."*

**References**

Azen, R. and Budescu, D. V.: Comparing predictors in multivariate regression models: An extension of dominance analysis, J. Educ. Behav. Stat., 31(2), 157–180, doi:10.3102/10769986031002157, 2006.

Budescu, D. V.: Dominance analysis: A new approach to the problem of relative importance of predictors in multiple regression, Psychol. Bull., 114(3), 542–551, doi:10.1037/0033-2909.114.3.542, 1993.

Thomas, B. F. and Famiglietti, J. S.: Identifying Climate-Induced Groundwater Depletion in GRACE Observations, Sci. Rep., 9(1), doi:10.1038/s41598-019-40155-y, 2019.

Tighe, E. L. and Schatschneider, C.: A dominance analysis approach to determining predictor importance in third, seventh, and tenth grade reading comprehension skills, Read. Writ., 27(1), 101–127, doi:10.1007/s11145-013-9435-6, 2014.

---

## Author Comment (AC3) · 22 Oct 2020

*"Importance of spatial and depth-dependent drivers in groundwater level modeling through machine learning"* by Pragnaditya Malakar, Abhijit Mukherjee, Soumendra N. Bhanja, Dipankar Saha, Ranjan Kumar Ray, Sudeshna Sarkar, Anwar Zahid

**Prof. Lahcen Benaabidate's Comment:**

The authors have treated an interesting topic dealing with groundwater in a large transboundary aquifer between India and Bangladesh for the purpose of highlighting the influence of various triggers; natural and anthropogenic that act and harm this groundwater. The investigation is carried out by the use of machine learning methods (support vector machines and artificial neural network The application of this kind of modeling constitutes a novelty for groundwater in the studied basin. The title is appropriate for the content of the paper, however, it will be better if they add an indication about the study area. The abstract summarizes the main information of the paper and highlights the main finding. The paper is well written and balanced. However, the article contains imperfections such as:

Reply: We thank Prof. Lahcen Benaabidate for his review and support for the general intent of the paper. We appreciate that the appended comments are helpful and intended to improve the manuscript. We have addressed the reviewer's comments, and we believe these have greatly improved the manuscript.

**Highlights of the revision:**

We have

  a) Provided the appropriate citations in the text and in the supplementary information.
  b) Addressed the high groundwater abstraction in southeast India (Bengal basin)
  c) Maintained an alphabetical order in the reference list
  d) Corrected the typos

**SC1. Comment 1:** The authors repeatedly cited Figures S1 up to S17 (line 263) and Tables S1 to S11 (example line 282, 283) but in the list of figures and tables below those illustrations are missing.

Reply: Thank you, Prof. Lahcen Benaabidate, for the comment. Figure S1 to S17 and Table S1 to S11 are supplementary figures and tables, which can be found in the supplementary information

section. Please find the supplementary information (https://hess.copernicus.org/preprints/hess-2020-208/hess-2020-208-supplement.pdf).

**SC1. Comment 2:** Line 47 they wrote: south-east India (Bengal basin), maybe they rather say North-east India.

Reply: We thank Prof. Benaabidate for the suggestions.

We would like to mention that in addition to severe groundwater depletion in northwest India, the Bengal basin in southeast India also suffered from pervasive groundwater abstraction in the recent past, reported by Mukherjee et al. (2007), Macdonald et al. (2015), Macdonald et al. (2016), Lapworth er al. (2018). Following the reviewer's suggestions, we have mentioned appropriate references in the text.

*"As a result of the pervasive groundwater withdrawals, IGBM experiences rapid groundwater depletion, predominantly in northwest India (Rodell et al., 2009), southeast India (Bengal basin) (Mukherjee et al., 2007, Macdonald et al., 2015, Macdonald et al., 2016, Lapworth er al., 2018), and the Meghna basin in Bangladesh (Shamsudduha et al., 2011; MacDonald et al., 2016)."*

**SC1. Comment 3:** Line 49, do "Summer" and "winter" correspond to "Rabi" and "Kharif" respectively.

Reply: We thank the reviewer for noticing the typo. Following the reviewer's comment, we have modified the text.

*"These densely populated agricultural regions of IGBM are dependent on the groundwater-fed irrigation for crop production, primarily for the summer season (i.e., Kharif) and winter season (i.e., Rabi) crops (World Bank, 2010)."*

**SC1. Comment 4:** Line 54, could you give value for the population?

Reply. Thank you. We have added the population and appropriate citation.

*"Thus, posing a severe threat to water sustainability for more than 1 billion people (Mukherjee, 2018) in South-Asia."*

**SC1. Comment 5:** Line 166, authors should add a reference.

Reply: Following Prof. Benaabidate's suggestion, we have mentioned appropriate reference in the text.

We added,

*"ANN is a data-driven computational method, which follows the biological neural system (Rajaee et al., 2019)."*

**SC1. Comment 6:** In all the manuscript, when a cardinal point is preceded by "the" the first letter should be written in capital letter.

Reply: Thank you for the comment.

**SC1. Comment 7:** Line 349, it's better to change "Please" » by "we".

Reply: Thank you, Prof. Benaabidate's. Following your suggestion, we have modified the sentence.

We modified,

*"We note that water use may not be strongly correlated to population, especially in rural areas, where pumping for irrigation is not necessarily linked to population."*

**SC1. Comment 8:** In the reference list, the citations should be written in the alphabetic order.

Reply: We thank Prof. Benaabidate for the comment.

Following the comment, we have maintained an alphabetical order in the reference list.

**SC1. Comment 9:** For some references with the same first author, they should be classified ..YEARa, YEARb…, example "Bhandri et al, 2019".

Reply: We thank Prof. Benaabidate for the comment.

Following the comment, we have maintained the order in the reference list.

**SC1. Comment 10:** There is a disagreement between some references in the test and in the list, example Youn et al, 2016 and in the list 2011. BADC, 2017 in the text and 2014 in the list.

Reply: Thank you, Prof. Benaabidate for the comment.

Yoon et al., 2011 and Yoon et al., 2016 both are cited in the text. Accordingly, they were added to the reference list.

Thank you for noticing the typo. Following the reviewer's comment, we have corrected the typo.

*"For Bangladesh, the groundwater withdrawals data were derived by integrating data from local and published datasets (AQUASTAT, 2018; Bangladesh Agricultural Development Corporation, 2014) (Table S3)."*

**References**

Bangladesh Agricultural Development Corporation. Minor Irrigation Survey report, 2013 – 14. Govt. of Bangladesh, Dhaka, 2014

**SC1. Comment 11:** The reference SEDAC, 2018 is missing n the list.

Reply: Thank you for the comment.

SEDEC is referenced under the recommended reference of the population data documentation, i.e., Palisades NY: NASA Socioeconomic Data and Applications Center (SEDAC)

which is,

Palisades NY: NASA Socioeconomic Data and Applications Center (SEDAC): Documentation for the Gridded Population of the 505 World, Version 4 (GPWv4), Revision 11 Data Sets, Cent. Int. Earth Sci. Inf. Netw. (CIESIN), Columbia Univ. 2018, III (845), 224–234, doi:https://doi.org/10.7927/H45Q4T5F, 2018.

**SC1. Comment 12:** The reference Bhanja et al, 2017 is missing in the list.

Reply: Thank you for noticing the typo. Following the reviewer's comment, we have corrected the typo in the text.

*"The uncontrolled irrigation practices (Barik et al., 2016; Bhanja et al., 2017a) lead to the over-exploitation of the aquifers in the IGBM, which is reflected in the deepening of GWL in northwest India, Meghna basin in Bangladesh, western Ganges basin and Bengal basin part of Ganges basin in the east (Figure 1b)."*

Reference

Bhanja, S. N., Mukherjee, A., Rodell, M., Wada, Y., Chattopadhyay, S., Velicogna, I., Pangaluru, K. and Famiglietti, J. S.: Groundwater rejuvenation in parts of India influenced by water-policy change implementation, Sci. Rep., 7(1), 7453, doi:10.1038/s41598-017-07058-2, 2017a.

**SC1. Comment 13:** In conclusion, I recommend that this paper will be accepted after minor revisions.

Reply: We thank Prof. Benaabidate for his appreciation of the manuscript.

---

## Author Comment (AC4) · 22 Oct 2020

*"Importance of spatial and depth-dependent drivers in groundwater level modeling through machine learning"* by Pragnaditya Malakar, Abhijit Mukherjee, Soumendra N. Bhanja, Dipankar Saha, Ranjan Kumar Ray, Sudeshna Sarkar, Anwar Zahid

**Prof. Saman Javadi's Comment:**

The authors have investigated the relative influence of major drivers in groundwater level change and linked them with the performance of machine learning-based predictive models, in a very important transboundary system. The study illustrates the advantages and limitations of machine learning-based modeling in a very heterogeneous regime. Due to this specific study area, this study is particularly important. The spatial and depth-dependent variability in model performance using GWL data is novel. The depth component of the study is particularly impressive, and probably first of its kind. In my view, the manuscript should be accepted with minor revision. The manuscript is well written, well-segmented and concise. However, there are some typos that should be corrected.

Reply: We thank Prof. Saman Javadi for his review and support for the general intent of the paper. The comments are very helpful and used to improve the manuscript while addressing them.

**Highlights of the revision:**

We have

a) Added a brief discussion on the major drivers of groundwater level change
b) Moved the flowchart to the main text
c) Described the hydrogeological conditions and aquifer characteristics and two maps added in this regard
d) Added the full forms of the abbreviation used in Figure 6
e) Explained the abbreviations in Table 1

**SC2. Comment 1:** If possible please add few lines on major drivers in the introduction section, importantly for the abstracted part of the aquifer.

Reply: We thank Prof. Javadi for the comment. Following Prof. Javadi's comment, we have added a brief description of major drivers on groundwater storage change.

We added,

*"There are disagreements in the researcher community on the major drivers influencing groundwater storage change over South Asia. Some studies have highlighted the significant relation between groundwater storage change and precipitations (Asoka et al., 2017, 2018). Other studies (Mukherjee et al., 2007; MacDonald et al., 2016; MacDonald et al., 2015; Bhanja et al., 2017a; Lapworth et al., 2018; Bhanja et al., 2019a; Bhanja et al., 2020) have indicated that groundwater abstraction (through influencing recharge, irrigation return flow) is the primary factor in groundwater storage change in the region."*

**Reference**

Asoka, A., Gleeson, T., Wada, Y. and Mishra, V.: Relative contribution of monsoon precipitation and pumping to changes in groundwater storage in India, Nat. Geosci., 10(2), 109–117, doi:10.1038/ngeo2869, 2017.

Asoka, A., Wada, Y., Fishman, R. and Mishra, V.: Strong linkage between precipitation intensity and monsoon season groundwater recharge in India, Geophys. Res. Lett., 45(11), 5536–5544, doi:10.1029/2018GL078466, 2018.

Bhanja, S. N., Mukherjee, A., Rodell, M., Wada, Y., Chattopadhyay, S., Velicogna, I., Pangaluru, K. and Famiglietti, J. S.: Groundwater rejuvenation in parts of India influenced by water-policy change implementation, Sci. Rep., 7(1), 7453, doi:10.1038/s41598-017-07058-2, 2017a.

Bhanja, S. N., Mukherjee, A., Rangarajan, R., Scanlon, B. R., Malakar, P. and Verma, S.: Long-term groundwater recharge rates across India by in situ measurements, Hydrol. Earth Syst. Sci., 23(2), 711–722, doi:10.5194/hess-23-711-2019, 2019a.

Bhanja, S. N., Mukherjee, A. and Rodell, M.: Groundwater storage change detection from in situ and GRACE-based estimates in major river basins across India, Hydrol. Sci. J., 65(4), 650–659, doi:10.1080/02626667.2020.1716238, 2020.

MacDonald, A.M., Bonsor, H. C., Taylor, R., Shamsudduha, M., Burgess, W. G., Ahmed, K. M., Mukherjee, A., Zahid, A., Lapworth, D., Gopal, K., Rao, M. S., Moench, M., Bricker, S. H., Yadav, S. K., Satyal, Y., Smith, L., Dixit, A., Bell, R., van Steenbergen, F., Basharat, M., Gohar, M. S., Tucker, J., Calow, R. . C. and Maurice, L.: Groundwater resources in the Indo-Gangetic Basin: resilience to climate change and abstraction, 2015.

MacDonald, A. M., Bonsor, H. C., Ahmed, K. M., Burgess, W. G., Basharat, M., Calow, R. C., Dixit, A., Foster, S. S. D., Gopal, K., Lapworth, D. J., Lark, R. M., Moench, M., Mukherjee, A., Rao, M. S., Shamsudduha, M., Smith, L., Taylor, R. G., Tucker, J., Van Steenbergen, F. and Yadav, S. K.: Groundwater quality and depletion in the Indo-Gangetic Basin mapped from in situ observations, Nat. Geosci., 9(10), 762–766, doi:10.1038/ngeo2791, 2016.

Lapworth, D. J., Zahid, A., Taylor, R. G., Burgess, W. G., Shamsudduha, M., Ahmed, K. M., Mukherjee, A., Gooddy, D. C., Chatterjee, D. and MacDonald, A. M.: Security of Deep Groundwater in the Coastal Bengal Basin Revealed by Tracers, Geophys. Res. Lett., 45(16), 8241–8252, doi:10.1029/2018GL078640, 2018.

**SC2. Comment 2:** It would be better if the flow chart is moved into the main article from the supplementary section.

Reply: Thank you, Prof. Javadi, for the comment. Following your comment, we have moved the flow chart in the main text.

**SC2. Comment 3:** The Geology and hydrology of the study area could be expanded a little more.

Reply: We thank the reviewer for his/her comment. The IGBM basin exhibits a wide range of permeability, transmissibility, hydraulic conductivity, and aquifer depth. The diverse depositional settings and environment of Pleistocene to Holocene sediments resulted in variable aquifer properties across the basin.  Following the reviewer's suggestions, we have added a brief description of the hydrogeological conditions and aquifer characteristics of the IGBM. Furthermore, we also added two figures showing the aquifer type, horizontal hydraulic conductivity, transmissivity, and specific yield of India and Bangladesh. We added,

"*The sediment (both recent Plio-Pleistocene to Holocene alluvium and older Miocene rocks) thickness of IGBM is up to 2 km (Singh et al., 1996). However, the effective thickness of the aquifer in most of the IGBM is generally the top 200 m. Notably, in the Bengal basin area in the eastern part of the Ganges basin and the Indus basin area, the effective aquifer thickness could be more than 300 m (Mukherjee et al., 2007; Macdonald et al., 2015). The diverse depositional setting and environment of Pleistocene to Holocene sediments resulted in variable aquifer properties across the basin (Bonsor et al., 2017). A distinct systematic reduction in permeability is found away from the mountain and towards the coast in most of the IGBM; however, the distribution is more*

*complex for the Ganges basin (Macdonald et al., 2015). The transmissivity within the upper and middle Ganges basin and most of the Brahmaputra basin ranges from several 100 $m^2day^{-1}$ to more than 5000 $m^2day^{-1}$ (Bonsor et al., 2017), which is representative of permeability values of $5 - 100$ m/d (CGWB 2010). However, in the Indus basin, the permeability values of $<10\ m.day^{-1}$ to $>60m$ $m^2day^{-1}$ is reported. Fig. S1 show the aquifer type, horizontal hydraulic conductivity, and transmissivity of India and Bangladesh (Bhanja et al., 2017a, 2019a). The specific yield in the unconsolidated sedimentary (high hydraulic conductivity) aquifer part of the IGBM ranges from 0.06 to 0.20 (mean 0.013). However, the specific yield values up to 0.08 are reported in the consolidated sedimentary (medium hydraulic conductivity) part of the basin (Bhanja et al., 2016). A specific yield map for India and Bangladesh is shown in Fig. S2."*

[Figure]

*Fig. S1. Different aquifer types, horizontal hydraulic conductivity (mday$^{-1}$) and transmissivity ($m^2day^{-1}$) for India and Bangladesh (modified from Bhanja et al., 2019a).*

[Figure]

*"Fig. S2. Specific yield map for India and Bangladesh (modified from Bhanja et al., 2016)."*

**Reference**

Bonsor, H. C., MacDonald, A. M., Ahmed, K. M., Burgess, W. G., Basharat, M., Calow, R. C., Dixit, A., Foster, S. S. D., Gopal, K., Lapworth, D. J., Moench, M., Mukherjee, A., Rao, M. S., Shamsudduha, M., Smith, L., Taylor, R. G., Tucker, J., van Steenbergen, F., Yadav, S. K. and Zahid, A.: Hydrogeological typologies of the Indo-Gangetic basin alluvial aquifer, South Asia, Hydrogeol. J., 25(5), 1377–1406, doi:10.1007/s10040-017-1550-z, 2017.

Bhanja, S. N., Mukherjee, A., Saha, D., Velicogna, I. and Famiglietti, J. S.: Validation of GRACE based groundwater storage anomaly using in-situ groundwater level measurements in India, J. Hydrol., 543, 729–738, doi:10.1016/j.jhydrol.2016.10.042, 2016.

Bhanja, S. N., Mukherjee, A., Rodell, M., Wada, Y., Chattopadhyay, S., Velicogna, I., Pangaluru, K. and Famiglietti, J. S.: Groundwater rejuvenation in parts of India influenced by water-policy change implementation, Sci. Rep., 7(1), 7453, doi:10.1038/s41598-017-07058-2, 2017a.

Bhanja, S. N., Mukherjee, A., Rangarajan, R., Scanlon, B. R., Malakar, P. and Verma, S.: Long-term groundwater recharge rates across India by in situ measurements, Hydrol. Earth Syst. Sci., 23(2), 711–722, doi:10.5194/hess-23-711-2019, 2019a.

Central Ground Water Board: Groundwater quality in shallow aquifers of India, Govt. of India, Ministry of Water Resources, Faridabad, 76 pp., 2010

MacDonald, A.M., Bonsor, H. C., Taylor, R., Shamsudduha, M., Burgess, W. G., Ahmed, K. M., Mukherjee, A., Zahid, A., Lapworth, D., Gopal, K., Rao, M. S., Moench, M., Bricker, S. H., Yadav, S. K., Satyal, Y., Smith, L., Dixit, A., Bell, R., van Steenbergen, F., Basharat, M., Gohar, M. S., Tucker, J., Calow, R. . C. and Maurice, L.: Groundwater resources in the Indo-Gangetic Basin: resilience to climate change and abstraction, 2015.

Mukherjee, A., Fryar, A. E. and Howell, P. D.: Regional hydrostratigraphy and groundwater flow modeling in the arsenic-affected areas of the western Bengal basin, West Bengal, India, Hydrogeol. J., 15(7), 1397–1418, doi:10.1007/s10040-007-0208-7, 2007.

Singh, I.B.: Geological evolution of Ganga Plain: an overview. J Palaeontol Soc India 41:99–137, 1996

**SC2. Comment 4:** In Figure 6 please mention the full form of the abbreviation used, at least in the figure captions.

Reply: We thank the reviewer for the comment.

We have added the full forms of the abbreviation used.

[Figure]

*"Figure 6. The relative contribution of the predictor variables on groundwater level variation, determined by the dominance analysis.*

*Abbreviations: IGBM, Indus-Ganges-Brahmaputra basin; I, Indus basin; G, Ganges basin; B, Brahmaputra basin; M, Meghna basin; ALL, all observation wells; SH, Shallow observation wells; DP, deeper observation wells; GWL, groundwater level; POP, population; GWW, groundwater withdrawals; PPTN, precipitation; TEMP, temperature; PET, potential evapotranspiration."*

**SC2. Comment 5:** In the ANN, SVM table (Table 1) the author should explain in short Model A, B, C.

Reply: The author would like to thank Prof. Javadi for the comment.

*"Abbreviations: ALL, all observation wells; SH, Shallow observation wells; DP, deeper observation wells; GWL, groundwater level; Model A, GWL as input; Model B, GWL + meteorological variables as input; Model C: meteorological variables as input."*